# Neurocognitive dynamics of preparatory and adaptive cognitive control: Insights from mass-univariate and multivariate pattern analysis of EEG data

José C. García Alanis[1]*, Malte R. Güth[1,2], Mira-Lynn Chavanon[1], Martin Peper[1]

1 Department of Psychology, Philipps-Universität Marburg, Marburg, Germany, 2 Center for Molecular and Behavioral Neuroscience, Rutgers University, New Brunswick, NJ, United States of America

* jose.alanis@staff.uni-marburg.de

**Data Availability Statement:** The data and code that support the findings of this study are openly available in on the OSF at http://doi.org/10.17605/OSF.IO/D49QW. The raw EEG data cannot be shared publicly because of data privacy

## Abstract

Cognitive control refers to humans' ability to willingly align thoughts and actions with internally represented goals. Research indicates that cognitive control is not one-dimensional but rather integrates multiple sub-processes to cope with task demands successfully. In particular, the dynamic interplay between preparatory (i.e., prior to goal-relevant events) and adaptive (i.e., in response to unexpected demands) recruitment of neural resources is believed to facilitate successful behavioural performance. However, whether preparatory and adaptive processes draw from independent or shared neural resources, and how these align in the information processing stream, remains unclear. To address these issues, we recorded electroencephalographic data from 52 subjects while they performed a computerised task. Using a combination of mass-univariate and multivariate pattern analysis procedures, we found that different types of control triggered distinct sequences of brain activation patterns, and that the order and temporal extent of these patterns were dictated by the type of control used by the participants. Stimuli that fostered preparatory recruitment of control evoked a sequence of transient occipital-parietal, sustained central-parietal, and sustained fronto-central responses. In contrast, stimuli that indicated the need for quick behavioural adjustments triggered a sequence of transient occipital-parietal, fronto-central, and central parietal responses. There was also a considerable degree of overlap in the temporal evolution of these brain activation patterns, with behavioural performance being mainly related to the magnitude of the central-parietal and fronto-central responses. Our results demonstrate how different neurocognitive mechanisms, such as early attentional allocation and subsequent behavioural selection processes, are likely to contribute to cognitive control. Moreover, our findings extend prior work by showing that these mechanisms are engaged (at least partly) in parallel, rather than independently of each other.

regulations. The data provided on the OSF are
anonymised (i.e., pre-processed) and epoched.

**Funding:** Research was supported in part by
grants awarded to José C. García Alanis by the
German Academic Exchange Service – DAAD
(grant denominations: STIBET, IPID4all). Open
access funding was provided by the Open Access
Publishing Fund of Philipps-Universität Marburg.

**Competing interests:** The authors have declared
that no competing interests exist.

# Introduction

Selecting and executing adequate responses to situational demands is crucial for effective daily life functioning. This ability is typically referred to as *cognitive control* [1, 2]. It is widely accepted that cognitive control is not one-dimensional. Rather, it is thought to rely on a set of basic, yet partially overlapping perception and reasoning-related processes to align thoughts and actions with goals [3–6]. These processes include mechanisms that help shift attention towards (potentially) relevant information [7], reduce distractions [8], and integrate prior knowledge, expectations, and feedback [see 9 for review] to improve ongoing performance or achieve better outcomes in future situations [cf. 1].

A growing body of research indicates that cognitive control is facilitated by the recruitment of resources across a neural network that spans occipital, parietal and frontal brain regions [10–12]. For example, transient (i.e., quick, momentary) activation of sensory (e.g., parietal and occipital cortex), recorded via non-invasive brain imaging methods (e.g., fMRI), have been used as a proxy for the engagement of lower-level neural mechanisms for identifying (and orienting to) contextual cues that support the preferential selection of a particular behaviour over other competing alternatives [11]. Conversely, sustained activation of frontal (e.g., dorsolateral prefrontal cortex) brain regions are thought to reflect the active maintenance of goal-representations in anticipation of goal-relevant events [13]. However, the interaction between these processes and their functional timing within the information processing stream remains unclear [1].

Due to its high temporal resolution, electroencephalography (EEG) is particularly well suited for interrogating the dynamic contributions of the different neural processes involved in preparatory and adaptive control recruitment [14–16]. In EEG research, particular attention has been paid to positive and negative polarisations of the Event-Related Potential (ERP) in choice reaction tasks [1]. For instance, the Contingent Negative Variation (CNV)—a sustained negative ERP observed at fronto-central electrodes before the onset of imperative stimuli [17] —is believed to reflect anticipatory attentional states [18] and preparatory recruitment of neural resources [19, 20]. Moreover, predictive stimuli (i.e., cues) have been shown to impact earlier ERP components [21], such as the N1—a negative parietal-occipital ERP that peaks 100–200 milliseconds (ms) post-stimulus [21] -, the P3b - a positive parietal ERP appearing around 300 ms post-stimulus [22]—and the central parietal positivity (CPP) [23]. The P3b and CPP share many spatio-temporal characteristics, but the CPP typically manifests later in time (approx. 500–700 ms from stimulus onset) [23]. These response patterns are believed to reflect enhanced sensory (e.g., cue identification) and behavioural configuration processes (e.g., stimulus-response mapping, pre-activating the adequate response) [23, 24], necessary for meeting (upcoming) behavioural demands. Additionally, similar late positive potential (LPP) modulations have been observed in other cue-related paradigms, such as fear conditioning [25]. These studies further support the idea that parietal positive amplitude responses play a crucial role in processing cues and preparing for subsequent actions.

While ERPs have been invaluable in identifying spatio-temporal regions of interest for understanding how different brain processes contribute to performance in complex cognitive control tasks, the accuracy of this approach is still debated [26–29]. Moreover, although investigations often use ERP response patterns as measures for (presumably) distinct neural processes [30–32], the ability of ERPs' to segregate different neurocognitive generators is limited. For example, the amplitude responses captured by the ERP typically evolve over broad time-scales without clear boundaries, complicating the precise tracking of the spatio-temporal evolution of neural activation and associated cognitive processes [29]. Furthermore, univariate statistical analyses of the ERP are capable of identifying time windows where information

processing differs between conditions, but they struggle to clearly distinguish between serial, parallel, or cascading activation of cognitive processes [33]. Indeed, ERP patterns, such as those seen during the P3b time-window, can arise from various cognitive architectures and multiple underlying neural generators [34, 35], but univariate analyses of the ERP can only approximate their order, overlap, or separability [36]. Lastly, there is a risk of missing crucial information if the focus of ERP analyses is restricted to a subset of channels or time windows that do not capture the full distribution of the effect in question [27], or if the effect's scalp distribution varies with stimuli or task properties [28].

## The present study

To address the limitations inherent in traditional ERP analyses and more accurately map the spatio-temporal evolution of EEG amplitude response patterns evoked during recruitment of cognitive control, we adopted a novel approach combining hierarchical mass univariate [37] and multivariate pattern analysis [MVPA; 33] To achieve this, we recorded continuous EEG while subjects performed the Dot Pattern Expectancy task [DPX: 38]. The DPX, an adaptation of the Continuous Performance Test (CPT) developed by Rosvold and colleagues [39], has been extensively used for cognitive control assessment [40–42]. The DPX, along with other variations, creates expectations for certain behavioural demands while also incorporating conditions that require overriding these expectations in response to changing behavioural requirements [43]. The goal is to separate behavioural preparation and anticipatory processes from mechanisms that enable a reactive adaptation to changes in behavioural demands [12, 44].

To our knowledge, only one additional study has analysed evoked amplitude responses in a continuous performance task with MVPA [45]. While this study primarily focused on discriminating individual conditions, it did not address the generalisability of the identified activity patterns across different stages of the information processing stream. In contrast, our research specifically aims to explore this important aspect. Nevertheless, the contribution by Sharifian and colleagues [45] to the cognitive control literature is significant, as they demonstrated that the effects of CPTs are broadly distributed across the scalp and not confined to individual ERPs.

The objective of the current study was twofold. First, we aimed to conduct a more detailed assessment of the amplitude response patterns evoked during the DPX and examined their association with behavioural performance. During the experiment, participants used the information provided by a cue-stimulus to prepare a response to a subsequent probe-stimulus. We expected that the preparatory recruitment of control would be beneficial for performance (i.e., faster responses and fewer errors) because, often in the paradigm, the prepared response was the one required upon the presentation of the probe. Preparatory recruitment of control is thought to depend on early attentional selection mechanisms [46]. Thus, we expected cues which more reliably predict subsequent demands to elicit an enhanced orienting response. We expected early attentional effects to manifest in heightened transient amplitude responses for highly predictive cues compared to more ambiguous cues. In addition, preparatory control recruitment is believed to heavily rely on another mechanism: The active maintenance of information (e.g., stimulus-response mappings) in working memory [12]. Consequently, we expect stronger sustained amplitude responses for highly predictive cues compared to more ambiguous ones.

During performance of the DPX, participants had to quickly adjust their behaviour if the probe-stimulus indicated that a response different from the one prepared upon presentation of the cue was necessary. In these situations, we expected an increased need for adaptive control to cope with the change in behavioural demands which hamper behavioural performance (i.e.,

slower responses and more errors [44]. Adaptive control is hypothesised to be facilitated by rapid, transient adjustments in the neural circuitry, triggered by changes in behavioural demands [6]. As such, we expect heightened transient amplitude responses for probes which signal the need for behavioural adaptation. These predictions align with previous research [46], and ERP components, such as the N1, P3b, and CNV provide appropriate priors for the spatio-temporal location of these effects [21, 47]. However, our study aimed to take a step further. By applying mass univariate analyses, we sought to account for the full spatio-temporal representation of the experimental effects. Our goal was to establish a more comprehensive foundation for future research and allow for a more mechanistic evaluation of evoked amplitude responses as neural markers for preparatory and adaptive control processes.

In addition, we used MVPA to build upon the results from the mass univariate analysis and provide further insights into the temporal evolution of the identified spatio-temporal patterns. By testing whether the identified spatio-temporal patterns are independent of each other or whether they share some degree of overlap in their functional timing, we can develop a more accurate model of the neurocognitive architecture underlying preparatory and adaptive control recruitment.

## Materials and methods

We report how we determined our sample size, all data exclusions, all manipulations, and all measures in the study [48].

### Statistical power

The required sample size was estimated using the "*pwr*" package [49] in the R-programming environment [50]. The necessary sample size to detect an effect size of at least d = 0.5 (medium effect size) with power = 0.80% at $p < 0.01$ in a paired-samples t-test (see statistical analysis, evoked amplitude response) was estimated to be approximately 50 subjects. We recruited additional subjects to account for potential data loss. The threshold for statistical significance was set to $p < 0.01$ for all analyses. All reported p-values are corrected for multiple comparisons using the methods described below.

### Participants

We recorded electroencephalographic data from 52 healthy young adults (30 females). Data were collected between 2016-07-18 and 2017-01-27. Subjects' age ranged between 18 and 29 years (M = 22.63, SD = 2.84). Female participants were slightly younger than male participants ($\Delta M = -1.88$, $t(50) = -2.50$, p = .016, $d = -0.71$, 99% $CI_d = [-1.45, 0.05]$). All subjects were undergraduate students at the Philipps-Universität Marburg and received course credit in exchange for participation. Subjects reported no current use of prescription drugs and no history of neurological or psychiatric disorders. All participants had normal or corrected-to-normal vision, assessed using a standardised Landolt C Test, and reported right-handed dominance. Participants provided written informed consent before participating in the experiment. The local ethics committee at the Department of Psychology of the Philipps-Universität Marburg approved the study before data collection.

### Experimental paradigm

Participants performed an adapted version of the DPX [41, 51]. The task was developed using Presentation® Software (Neurobehavioral Systems, Inc., Berkeley, CA, www.neurobs.com)

and was displayed using a BenQ XL2430T monitor (resolution: 1920x1080 pixels; refresh rate 144 Hz).

In the DPX, participants must discriminate between valid and non-valid pairs of cues and probes. As illustrated in Fig 1, participants were instructed to perform a target response (press the left button on) when a specific probe (referred to as the "X" probe) appeared after a specific cue (referred to as the "A" cue). For all other pairs in which the cue was not "A" (collectively referred to as "B" cues) or the probe was not "X" (collectively referred to as "Y" probes), participants should perform a non-target response. Participants responded by pressing a button with either the index (target) or middle finger (non-target) of the right hand on a Cedrus RB-840 response pad (Cedrus Corporation, San Pedro, CA). It is important to note that participants only responded after the probe was presented. Thus, to perform well in the task, participants needed to decode the information provided by the cue (i.e., the first stimulus) and maintain this information in working memory until the probe appeared on the screen. Then, they needed to assess whether both stimuli conformed to a valid cue-probe combination.

Participants first performed a brief practice run of 18 trials (50% "AX") to familiarise themselves with the task. In the main experiment, "AX" pairs comprised approximately 70% of all cue-probe pairs. This manipulation was introduced to induce a strong expectation that an "X" probe following an "A" cue making the "AX" pair the most common and expected condition). Conversely, the "AY", "BX", and "BY" pairs each accounted for roughly 10% of the trials and were therefore significantly less frequent. "AY" trials involved a predictive "A" cue followed by an unexpected "Y" probe, requiring participants to override the strong expectation and

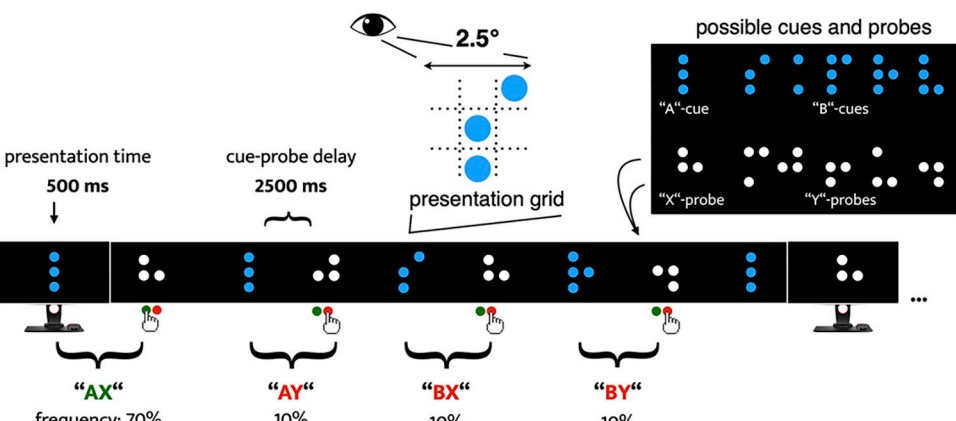

**Fig 1. Schematic illustration of the Dot Pattern Expectancy (DPX) task.** *Note*. Participants were presented with a continuous stream of dot patterns. All dot patterns fitted within a grid of 2.5˚ x 2.5˚ viewing angle, with single dots appearing at one of nine different positions. Presentation of the dot patterns followed the same standard procedure throughout the task. A cue stimulus (white dot pattern) appeared on the screen for 500 milliseconds (ms), an inter-stimulus interval (ISI; an empty black screen) of 2000 ms followed (i.e., stimulus-onset-asynchrony = 2500 ms), and then a probe stimulus (blue dot pattern) appeared for 500 ms. After the probe, a variable inter-trial interval (ITI) of 2500 to 3500 ms followed, and a new cue-probe sequence started. The inset (upper right corner) shows the set of dot patterns that could appear as cues or probes. Participants were instructed to perform a target response every time an "X" probe was presented following an "A" cue (i.e., after each "AX" pair). For all other cue-probe pairs ("AY", "BX", and "BY"; where "B" cues are any letter other than "A", and "Y" probes are any letter other than "X") participants should perform a non-target response. "AX" pairs were the most frequent in the task, creating a prepotent tendency to perform target responses. Namely, in "A"-cue trials, participants could prepare the target response (a subsequent "X" probe was more probable than a "Y" probe), but needed to adjust their behaviour accordingly if a "Y"-probe was presented. In "B"-cue trials, the cue provided all information needed to select the adequate response. This should allow participants to engage in response preparation without uncertainty. However, some behavioural interference could still arise in "BX" trials, as "X" probes are most often paired with "A" cues (i.e., interference between the prepared non-target response and the more often required target response).

perform a non-target response–this is a hallmark of reactive cognitive control. In contrast, "BX" and "BY" trials involved an unambiguous and highly preparatory "B" cue, as this cue provided all the information needed for the selection of the correct response, highlighting proactive control.

The experiment was divided into two blocks, each comprising 129 cue-probe pairs (90 "AX", 13 "AY", 13 "BX", and 13 "BY" pairs). A short break was allowed between blocks. To mitigate potential unsystematic variations (e.g., pronounced eye movements during stimulus presentation, head movements) and reduce stimulus input to the essential minimum, we adapted the DPX as follows. First, a distance of one metre between the participant and the monitor was kept as consistent as possible across participants. Second, all dot patterns were presented within a 2.5 x 2.5-degree viewing angle grid. Single dots appeared at one of nine possible positions, enhancing the consistency of sensory input from both the cue and probe stimuli. Third, during both the Inter-stimulus Interval (ISI) and the Inter-trial Interval (ITI), which were of similar duration, a blank screen was displayed. Consequently, participants saw a continuous stream of stimuli, rather than individual, distinct "trials". This represents a departure from other versions of the DPX [22, 42]. However, we believe these modifications accentuate the continuous performance aspect of the task, underlining the necessity for vigilance [52] and the retention of relevant information in working memory until needed.

## EEG data acquisition and pre-processing

Continuous EEG data were recorded using a BioSemi ActiveTwo system (BioSemi, Amsterdam, Netherlands), with 64 silver/silver-chloride active scalp electrodes, positioned according to the international 10–20 standard [53]. Data were digitised using ActiView software (V6.05, BioSemi, Amsterdam, Netherlands) at a sampling rate of 512 Hz (2048 Hz at 1/4 decimation rate). Eye movements were recorded using two pairs of flat electrodes (electrooculogram, EOG). One pair was placed on the supra- and infra-orbital rims of the right eye, and the other pair on the outer canthi of the eyes. During recording, the EEG signal was referenced to two linked electrodes—the Common Mode Sense (active) and the Driven Right Leg (passive)—positioned midway between the POz and PO3/PO4 locations.

EEG data were processed offline using MNE-Python 1.5.1 [54] and custom scripts written for Python 3.11. First, data were screened for noisy channels using a fully automated robust z-scoring procedure [55]. Channels were identified as noisy if they exhibited abnormal amplitude levels (high or low compared to the median channel amplitude), poor correlation with other channels (correlation < 0.4 for over 1% of the experiment), or contained excessive high-frequency noise (> 50 Hz) [56]. Noisy channels were interpolated using spherical splines [57]. Following, data were re-referenced to the average of all 64 channels. Line noise was removed using a 50 Hz notch filter (0.5 Hz upper and lower transition bandwidth). Next, we applied a one-pass, zero-phase, non-causal band-pass filter (windowed time-domain design method; hamming window with 0.0194 passband ripple and 53 dB stopband attenuation; lower passband edge at 0.10 Hz; lower transition bandwidth: 0.10 Hz, -6 dB cutoff frequency: 0.01 Hz; upper passband edge: 30.00 Hz, upper transition bandwidth: 7.50 Hz, -6 dB cutoff frequency: 33.75 Hz) to remove slow drifts and remaining high frequency noise.

To correct for stereotypical signal artefacts, such as those caused by ocular movements, we created a 1 Hz high-pass filtered copy of the EEG data and decomposed it into independent components using extended infomax independent component analysis [58]. Using the IC-label algorithm [59, 60], we identified components that closely matched oculomotor and muscle activity and removed them. We used the remaining components to remix the EEG data. Lastly, we segmented the EEG data into single epochs centred around stimulus presentation

(For mass univariate analyses: -500 to 2500 ms around cue presentation and -500 to 1000 ms around probe presentation; For MVPA: -500 to 3500 ms around cue presentation). Any epochs that still contained signal artefacts were automatically excluded using the *Autoreject* algorithm (version 0.4.0) with default settings [61]. The preprocessing pipeline produced an. html preprocessing report for each subject for visual inspection and quality control. These reports are provided on the projects OSF website.

## Statistical analyses

**Behavioural performance.** Participants performance in the DPX was analysed using a hierarchical, two-level linear mixed-effects regression approach. Hierarchical regression models allow for the examination of relationships between predictor and response variables (fixed effects), while accounting for individual differences in the distributional characteristics of the response variable (random effects). Before analysis, we excluded extreme response times (i.e., RT < 100 ms and > 1000 ms). Categorical variables were effect (i.e., deviation) coded, and all continuous variables grand-mean centred around a mean of zero. Participants' age and gender were introduced as second-level covariates to account for individual differences in these variables. To address potential issues arising from interdependencies among measurements and varied numbers of measures (analysis focused on single trial data) per condition and participant, all models estimated by-subject random effects by nesting data within participants.

We used the "*lme4*" [62] and "*lmerTest*" [63] packages in the R-programming environment to fit all behavioural performance models by Restricted Maximum Likelihood. Effect sizes were estimated using Cohens' d. Additional resources used for data processing and visualisation include "*emmeans*", "*effectsize*", "*performance*", "*gt*", "*dplyr*", "*ggplot2*", and "*papaja*" [64–72].

**Evoked amplitude response.** *Mass univariate analysis.* Similar to behavioural data, we analysed amplitude responses to cue and probe stimuli using a hierarchical, two-level mass univariate regression analysis approach [37]. We transformed the segmented EEG data of each participant (a three-dimensional matrix of segments—time locked to the onset of cues or probes -, channels, and time samples) into a two-dimensional space via matrix vectorisation. This procedure resulted in a matrix **Y**, where each row corresponds to one EEG segment (**m** rows in total), and columns (**n** columns in total) that contain the amplitude values recorded at each time sample at electrode, concatenated in sequence. To link each row of the **Y** matrix to an experimental condition, we used a design matrix **X**, comprising **m** rows (one for each EEG segment) and **k** columns (each representing a contrast coded predictor for the experimental conditions, e.g., predictive A cue = -1, highly proactive B cue = 1).

Level 1 analysis consisted of solving the least-squares problem for the linear system $\mathbf{Y} = \mathbf{X}\boldsymbol{\beta} + \boldsymbol{\varepsilon}$ (a general linear model). This allowed us to estimate the relationship between **Y** (the data) and **X** (the experimental conditions) at the subject level. For each participant, we computed the coefficient matrix $\boldsymbol{\beta}$ ($\mathbf{k} \times \mathbf{n}$) in a mass univariate manner (i.e., one $\boldsymbol{\beta}$-weight for each channel and each time sample) by solving $\boldsymbol{\beta} = \mathbf{X}^{+}\mathbf{Y}$. Here, $\mathbf{X}^{+}$ is the generalised Moore-Penrose pseudo-inverse [cf. 37] of the design matrix **X**, computed using singular value decomposition ($\mathbf{X}^{+} = \mathbf{V}\boldsymbol{\Sigma}^{+}\mathbf{U}^{*}$, where $\mathbf{X} = \mathbf{U}\boldsymbol{\Sigma}\mathbf{V}^{*}$), as implemented in the Python package SciPy [73].

For level 2 analysis, we compared the estimated condition-specific $\boldsymbol{\beta}$-weights (i.e., the regression coefficients) across subjects to assess statistical significance. For this purpose, coefficients were contrasted against each other via a paired samples t-test. To assess the significance of each contrast and account for multiple comparisons, we employed a bootstrap procedure with F-max statistic thresholds, as described by Pernet and colleagues [37, 74]. First, we calculated the observed t-values for each pairwise contrast (e.g., predictive "A" vs. highly proactive

"B" cues) using a paired-samples t-test. Second, we grand-mean centred the condition-specific regression coefficients around zero and randomly sampled subjects (with replacement) to create a new bootstrap sample. Third, for each bootstrap sample, we performed another paired-samples t-test, computed the corresponding F-value and saved the maximum test statistic. The second and third steps were repeated 2000 times. The resulting distribution of maximum test statistics provided an approximation of the true effect under the null hypothesis (H0). This distribution was used to determine critical test values corresponding to a significance level of p = 0.01 (we used a more conservative threshold of p = 0.01 as opposed to the typically used p = 0.05 due to the high number of tests performed during this analysis step). Effects were considered significant if the test values from the observed sample's spatio-temporal t-test exceeded the 99% quantile of the maximum test statistics distribution. This approach effectively and conservatively controls for multiple testing (similar to Bonferroni correction) and has the advantage of having an exact Type 1 error rate, as the H0 distribution accounts for all tests performed (across all channels and time samples) and is not tied to a specific location in space and time.

*Multivariate pattern analysis (MVPA).* We used MVPA to find patterns of EEG activity (i.e., linear combinations of amplitude values recorded at multiple channels) that best discriminated between experimental conditions. To this end, we trained linear Support Vector Classifiers (SVC; cf.) [75] to predict whether a predictive "A" cue or a highly proactive "B" cue was presented or whether an expected "X" probe or an unexpected "Y" probe was presented after a cue "A". To account for the imbalance of the experimental paradigm (e.g., "AX" cues-probe pairs are more frequent than "AY" pairs), we used a regularisation parameter of **C = 1 × condition weight**, with the condition weights set inversely proportional to the frequency of the conditions in each participant's data set.

The classifiers were trained independently for each participant and at each time sample of the analysis time window (-500 to 3500 ms around cue presentation, thus including 1000 ms of probe stimulus processing time, i.e., from 2500 to 3500 ms). We used a 5-fold stratified cross-validation procedure to assess the performance of these classifiers. For this purpose, we divided each participant's single-trial EEG data into five subsets, or "folds". For each trial in each fold, we used a classifier—trained on the other four folds—to predict the experimental condition at each time sample, thus providing a binary prediction for each time sample.

The accuracy of the classifiers was computed by calculating the area under the Receiver Operating Characteristic curve for each fold and averaging across folds. These accuracy scores range from 0 to 1 and can be seen as non-parametric measures of effect size that compare true positive to false positive classifications [76]. A score of 1 signifies perfect accuracy, while 0.5 denotes chance level. High scores indicate a strong differentiation in the evoked amplitude response pattern between conditions. In terms of signal-detection theory, classifier accuracy can be interpreted as the strength of the difference between conditions in the underlying neural pattern at that given point in time [33, 77].

To analyse sequence effects during stimulus-evoked processes, we employed a generalisation across time (GAT) approach, as proposed by King and Dehaene [33]. Classifier performance was not only tested at the time sample that it was trained on, but also at every other time sample of the analysis time-window. This resulted in a classification matrix, or GAT matrix, with training times on the y-axis and testing times on the x-axis. The matrix diagonal represents analogous training and testing time points (i.e., when the classifier was trained at 200 ms post-stimulus and tested at the same time sample). Off-diagonal performance indicates pattern persistence or recurrence (i.e., time samples where a classifier trained at time sample **t** can still successfully predict the single-trial condition at time sample **t′**). This is indicative of

overlapping patterns of brain activity, and by extension overlapping engagement of cognitive processes [33, 36].

For assessing statistical significance, we compared classifier performance against chance level using a one-sample t-test with p < 0.01. To correct for multiple comparisons, we applied the Bonferroni method. All tests were two-sided. We performed MPVA analysis based on the python code provided by Heikel et. al [29]. The complete analysis scripts needed to reproduce the results are provided on GitHub (https://github.com/JoseAlanis/eeg_patterns_dpx).

## Results

### Behavioural performance

**Response time.** Response time (RT) was analysed using a two-level hierarchical mixed effects regression approach. The model estimated RT in the DPX, using accuracy (correct vs incorrect response), cue type ("A" vs "B"), and probe type ("X" vs "Y") as fixed within-subjects predictors. Further, the model estimated by-subjects random effects for the interaction between cue and probe, nesting observations within participants. The model achieved substantial explanatory performance with an $R^2_{conditional}$ = 0.508, and the proportion of variance explained by the fixed effects was $R^2_{marginal}$ = 0.240. The model's intercept (the overall mean RT) was estimated to be 420 ms (99% CI = [401, 440]). S1 Table in S1 File summarises the omnibus test results for all the predictors.

The model estimated a significant three-way interaction between accuracy, cue type, and probe type (F(1,124) = 69.110, p < 0.001). Notably, the model estimated faster RTs for correct "AX" responses than correct "AY" responses ($\Delta M_{AY -AX}$ = 140 ms, 99% CI = [105, 176], $d$ = 2.18, 99% $CI_d$ = [1.50, 2.87], p < 0.001; see Fig 2a, left panel). This effect is consistent with our expectations, as we hypothesised that accurate performance in "AY" trials would require additional recruitment of reactive control resources to override the default behaviour associated with "A" cues. Importantly, the model estimated comparable effects across participants (see Fig 2A), indicating that "AY" trials systematically increased the control demands required for correct performance. This effect was corroborated by a reversal of the experimental effect

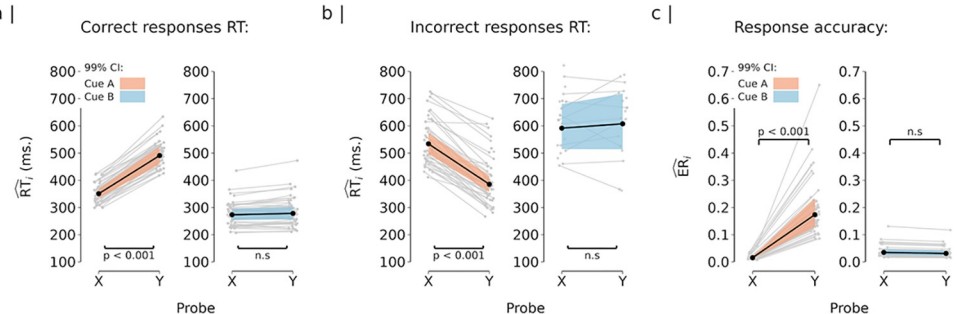

**Fig 2. Behavioural performance during the DPX task.** *Note.* Depicted are the estimated marginal means (i.e., the model's predictions) obtained through linear mixed effect regression analysis of response times (RT) and error rates (ER) data. Solid lines depict the estimated population-level effect. The coloured shading surrounding the solid lines expresses the uncertainty of the effects with a 99% confidence interval (CI). The grey lines in the background depict the subject-level estimates. Estimated RT is provided in milliseconds (ms) and estimated errors rates are provided as proportion (scale 0.0 to 1.0). Panel a) depicts the estimated RT for correct responses. Panel b) depicts the estimated RT for incorrect responses. Correct RT in AX (M = 351 ms, CI = [334, 369]) were faster than correct RT in "AY" trials (M = 491 ms, CI = [460, 524]). In contrast, "BX" (M = 273 ms, CI = [254, 294]) and "BY" trials (M = 278 ms, CI = [256, 302]) elicited comparable correct RT. (b) Incorrect RT in "AY" (M = 385 ms, CI = [357, 416]) was faster than in "AX" trials (M = 534 ms, CI = [497, 573]). "BX" (M = 592 ms, CI = [515, 680]) and "BY" trials (M = 608 ms, CI = [514, 718]) showed comparable incorrect RT. Panel c) depicts the response accuracy, estimated ER are provided on a scale from 0 to 1. Error rates were higher in "AY" (M = 0.17, CI = [0.13, 0.23]) compared to "AX" trials (M = 0.02, CI = [0.01, 0.02]). "BX" (M = 0.04, CI = [0.03, 0.05]) and "BY" showed similar error rates (M = 0.03, CI = [0.02, 0.04]).

for incorrect responses ($\Delta M_{AY\text{-}AX}$ = −149 ms, 99% CI = [−202, −095], $d$ = −1.04, 99% CI$_d$ = [−1.36, −0.72], p < 0.001; see Fig 2B left panel).

Conversely, the model estimated similar RT for correct responses following both "BX" and "BY" trials ($\Delta M_{BY-BX}$ = 5 ms, 99% CI = [−13, 23], $d$ = 0.18, 99% CI$_d$ = [−0.24, 0.60], p = 1.000; see Fig 2A). Similarly, there was no difference in RT between incorrect "BX" and "BY" trials (see Fig 2B, right panel). This finding is in line with our expectations, as highly proactive "B" cues provided all the information necessary to prepare the correct response and the probe had no effect on the required behaviour–a hallmark of proactive cognitive control. Importantly, "BX" and "BY" trials showed faster RT compared to "AX" trials ($\Delta M_{AX-BX}$ = 77 ms, 99% CI = [55, 100], $d$ = 2.11, 99% CI$_d$ = [1.39, 2.83], p < 0.001; $\Delta M_{AX-BY}$ = 72 ms, CI99% = [45, 99], $d$ = 1.70, 99% CI$_d$ = [1.05, 2.36], p < 0.001). This indicates that participants efficiently engaged preparatory control in "B"-cue trials, further corroborating the cue's informational value in guiding anticipatory response selection.

**Response accuracy.**   Error rates were analysed using a two-level hierarchical mixed effects regression. The model estimated error rates in the DPX using cue type ("A" vs "B") and probe type ("X" vs "Y") as fixed within-subjects predictors. Further, the model estimated by-subjects random effects (i.e., nesting observations within participants). The model achieved substantial explanatory performance with an $R^2_{conditional}$ = 0.699, and the proportion of variance explained by the fixed effects was $R^2_{marginal}$ = 0.551. The model's intercept (the overall mean error rate) was estimated to be 0.042 (99% CI = [0.034, 0.051]). S2 Table in S1 File summarises the omnibus test results for all the predictors.

The model revealed a significant two-way interaction between cue and probe type (F(1,153) = 186.685, p < 0.001). It predicted higher error rates for "AY" trials compared to "AX" trials ($\Delta M_{AY-AX}$ = 0.158, 99% CI = [0.097, 0.219], $_d$ = 0.68, 99% CI$_d$ = [0.44, 0.92], p < 0.001; see Fig 2C, left panel). In contrast, the model estimated comparable error rates for "BX" and "BY" trials ($\Delta M_{BY-BX}$ = −0.004, 99% CI = [−0.018, 0.010], $d$ = −0.07, 99% CI$_d$ = [−0.28, 0.14], p = 1.000, see Fig 2C, right panel), but significantly lower error rates for "BX" and "BY" trials compared to "AX" trials ($\Delta M_{AX-BX}$ = −0.019, 99% CI = [−0.031, −0.007], $d$ = −0.42, 99% CI$_d$ = [−0.65, −0.20], p < 0.001; $\Delta M_{AX-BY}$ = −0.016, 99% CI = [−0.027, −0.005], $d$ = −0.38, 99% CI$_d$ = [−0.16, −0.60], p < 0.001). These results are in line with our predictions and provide further support to the findings from the RT analysis.

**Evoked amplitude response.**   We analysed evoked amplitude response patterns using a two-level hierarchical mass univariate analysis approach. Condition effects were estimated on single-trial data from each subject and then compared across subjects to assess significance (see Methods, mass univariate analysis).

## Cue stimuli

As depicted in Fig 3, analysis of the cue evoked activity revealed significant differences between responses elicited by the cue stimuli (i.e., predictive "A" cues and highly proactive "B" cues). The earliest differences emerged at approximately 180 to 250 ms post-cue. In this time window, "B" cues elicited a stronger negative polarisation of the evoked amplitude response at parietal and occipital channels compared to "A" cues. The peak of this effect was observed at channel P6 at approximately 219 ms post-stimulus ($t(51)$ = −7.19, $d$ = −1.01, 99% CI$_d$ [−1.45, −0.56]); see Fig 4A and 4D). "B" cues also showed a more pronounced positive amplitude response at frontal channels during the same time window compared to "A" cues (peak at channel AF7 at approximately 211 ms post-cue, $t(51)$ = 7.41, $d$ = 1.04, 99% CI$_d$ [0.59, 1.48]).

A second cluster of differences extended from approximately 400 to 750 ms post-cue. During this time window, "B" cues showed a stronger positive polarisation of the evoked

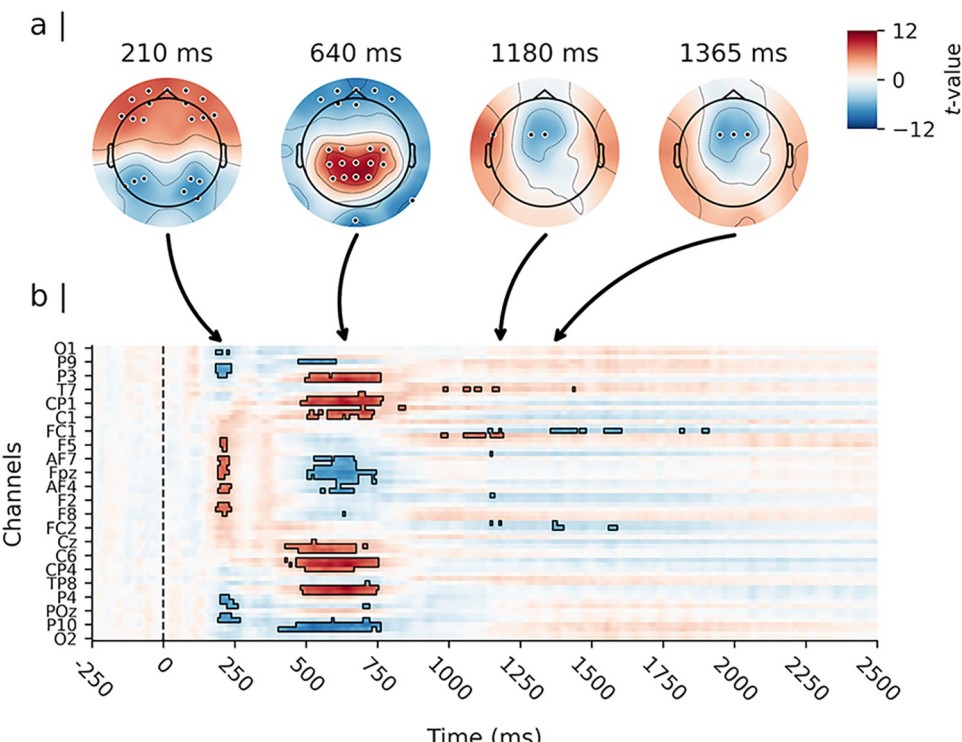

**Fig 3. Results of the mass univariate analysis of the cue-evoked amplitude response.** *Note.* Depicted are the results of a paired-samples t-test contrasting cue "B" minus cue "A". Panel a) shows the topographical distribution of the estimated effect (i.e., the t-values) on the scalp for four representative spatio-temporal clusters. Significant channels (p<0.01) are highlighted. Panel b) shows the estimated effects for the complete cue retention interval and the entire channel space, time locked to the onset of the cue. Time is depicted on the X-axis in b), with axis-ticks appearing every 250 milliseconds (ms). Time 0 ms depicts the onset of the cue stimulus. Only every 3rd channel is labelled on the Y-axis in b) to avoid clutter. The highlighted time samples in b) are significant at p<0.01, corrected for multiple comparisons.

amplitude response at parietal channels compared to "A" cues. The peak of this effect was observed at channel Pz at approximately 641 ms post-stimulus ($t(51) = 10.93$, $d = 1.53$, 99% $CI_d$ [1.00, 2.06]; see Fig 4B and 4E). "B" cues also showed a more pronounced negative response at occipital channels during the same time window compared to "A" cues. This effect reached its peak at channel Iz at approximately 523 ms post-cue ($t(51) = -9.45$, $d = -1.32$, 99% $CI_d$ [−1.82, −0.83]).

Finally, the analysis revealed two smaller clusters of differences at approximately 1148 ms and 1375 ms post-cue. During these time windows, "B" cues elicited a more negative amplitude response at fronto-central channels compared to "A" cues at channel FC1 ($t(51) = -6.86$, $d = -0.96$, 99% $CI_d$ [−1.39, −0.52]; see Fig 4C and 4F).

## Probe stimuli

Similarly, we found significant differences in the amplitude responses elicited by "X" and "Y" probes when they were presented after an "A" cue (i.e., between "AX" and "AY" cue-probe pairs). As shown in Fig 5, the earliest differences emerged at approximately 180 to 250 ms post-probe. In this time window, "Y" probes showed a stronger negative polarisation of the evoked amplitude response at parietal and occipital channels compared to "X" probes. The peak of this effect occurred at channel PO4 at approximately 219 ms post-stimulus ($t(51) =$

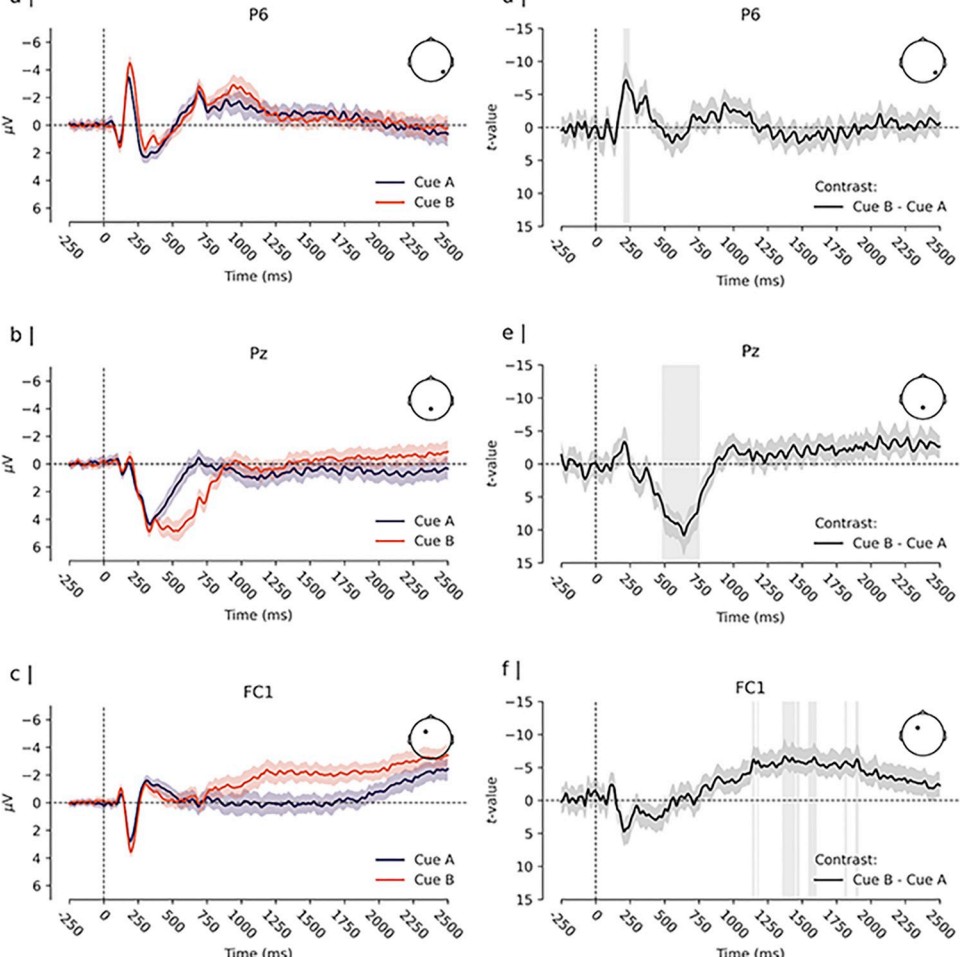

**Fig 4. Cue-evoked ERPs and estimated difference.** *Note.* Panels a), b), and c) depict the Event-Related-Potential (ERP, i.e., the grand average of the participants' mean evoked amplitude response, time locked to the onset of the cue) for cue "A" and cue "B". Time 0 ms depicts the onset of the cue stimulus. Depicted are channels that were located within clusters of significant differences between cue "A" and cue "B". The solid lines depict the ERP for each condition. The coloured shades surrounding the solid lines express the uncertainty of the estimates, with a 99% within-subjects confidence interval [cf. 78]. Panels d), e), and f) depict the time course of the estimated difference effect at each corresponding channel. The solid line depicts the t-value for the contrast "B" cues—"A" cues, and the shaded region surrounding the line expresses its uncertainty with a 99% confidence interval. Vertical grey bars highlight times of significant difference between conditions.

−7.43, $d = −1.04$, 99% $CI_d$ [−1.49, −0.59], see Fig 6A and 6D). Moreover, "Y" probes elicited a more pronounced positive amplitude response at left frontal channels compared to "X" probes during the same time window. The peak of this response was observed at channel FC5 at approximately 227 ms post-stimulus ($t(51) = 7.45$, $d = 1.04$, 99% $CI_d$ [0.59, 1.49]).

A second cluster of differences spanned from approximately 300 to 400 ms following the probe presentation. During this period, "Y" probes elicited a stronger positive polarisation of the evoked amplitude response at fronto-central channels. The peak of this effect was observed at channel FCz at approximately 367 ms post-stimulus ($t(51) = 7.93$, $d = 1.11$, 99% CI$d$ [0.65, 1.57], see Fig 6B and 6E). "Y" probes also elicited a more pronounced negative response at left occipital channels (peak at Iz at approximately 398 ms) compared to "X" probes during the same time window ($t(51) = −7.45$, $d = −1.04$, 99% $CI_d$ [−1.49, −0.59]).

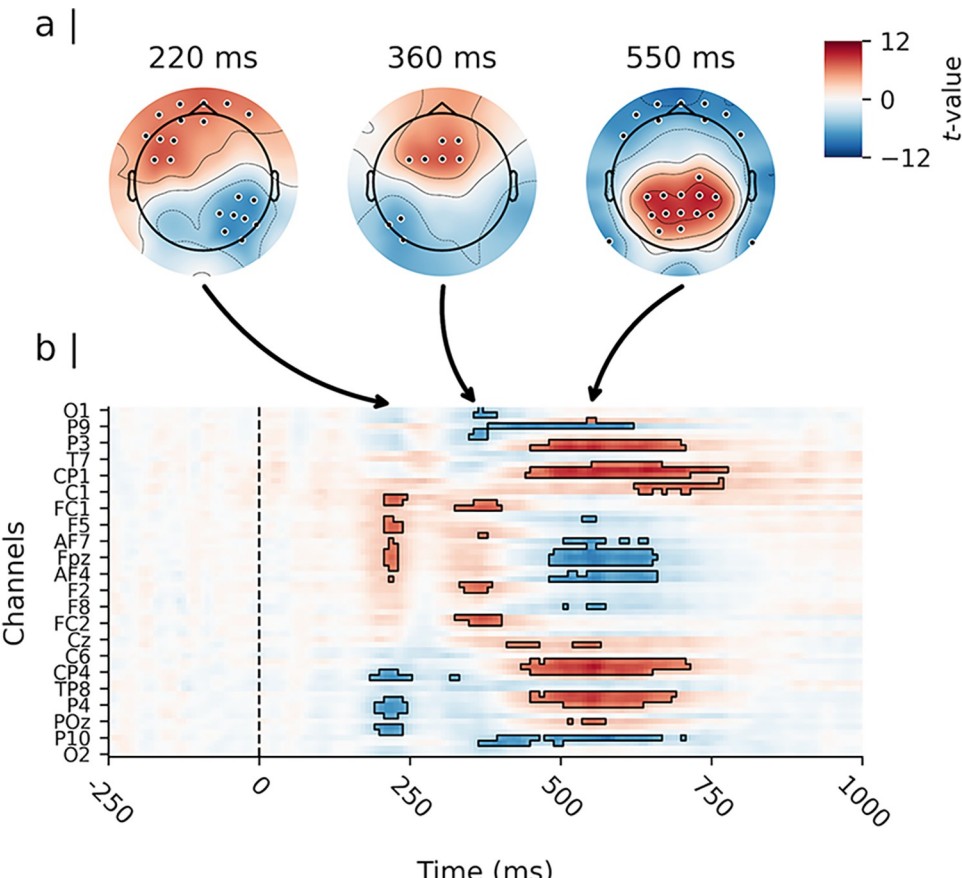

**Fig 5. Results of the mass univariate analysis of the probe-evoked amplitude response for the contrast "AY" minus "AX".** *Note.* Depicted are the results of a paired-samples t-test contrasting "Y" probes minus "X" probes in "A" cue trails (i.e., the contrast of probe evoked activity between "AY" and "AX" pairs, time locked to probe onset). Panel a) shows the topographical distribution of the estimated effect (i.e., the t-values) on the scalp for three representative spatio-temporal clusters. Significant channels (p<0.01) are highlighted. Panel b) shows the estimated effects for the complete probe processing interval and the entire channel space. Time is depicted on the X-axis in b), with axis-ticks appearing every 250 milliseconds (ms). Time 0 ms depicts the onset of the probe stimulus. Only every 3rd channel is labelled on the Y-axis in b) to avoid clutter. The highlighted time samples in b) are significant at p<0.01, corrected for multiple comparisons.

The analysis revealed a third cluster of differences spanning from approximately 450 to 750 ms after the probe presentation. Within this time window, "Y" probes exhibited a more positive amplitude response at parietal channels. The peak of this effect was observed at CP2 at approximately 555 ms ($t(51) = 9.70$, $d = 1.36$, 99% $CI_d$ [0.86, 1.86], see Fig 6C and 6F).

Finally, we found a small cluster of differences in the probe-evoked amplitude response between "AX" and "BX" probes (more positive response for "AX" probes at 150 ms at parietal channels; $t(51) = 6.21$, $d = 0.87$ $CI_d = [0.44, 1.29]$) and no significant differences "BX" and "BY" probes. For the sake of conciseness, these results are elaborated on in S1 and S2 Figs in S1 File.

**Functional timing of amplitude response patterns evoked by preparatory and adaptive control.** We used MVPA to further elucidate the temporal evolution of the identified amplitude response patterns [33]. We trained multivariate pattern classifiers to distinguish between experimental conditions based on evoked amplitude patterns and tested their ability to discriminate between conditions across different stages of the information processing stream.

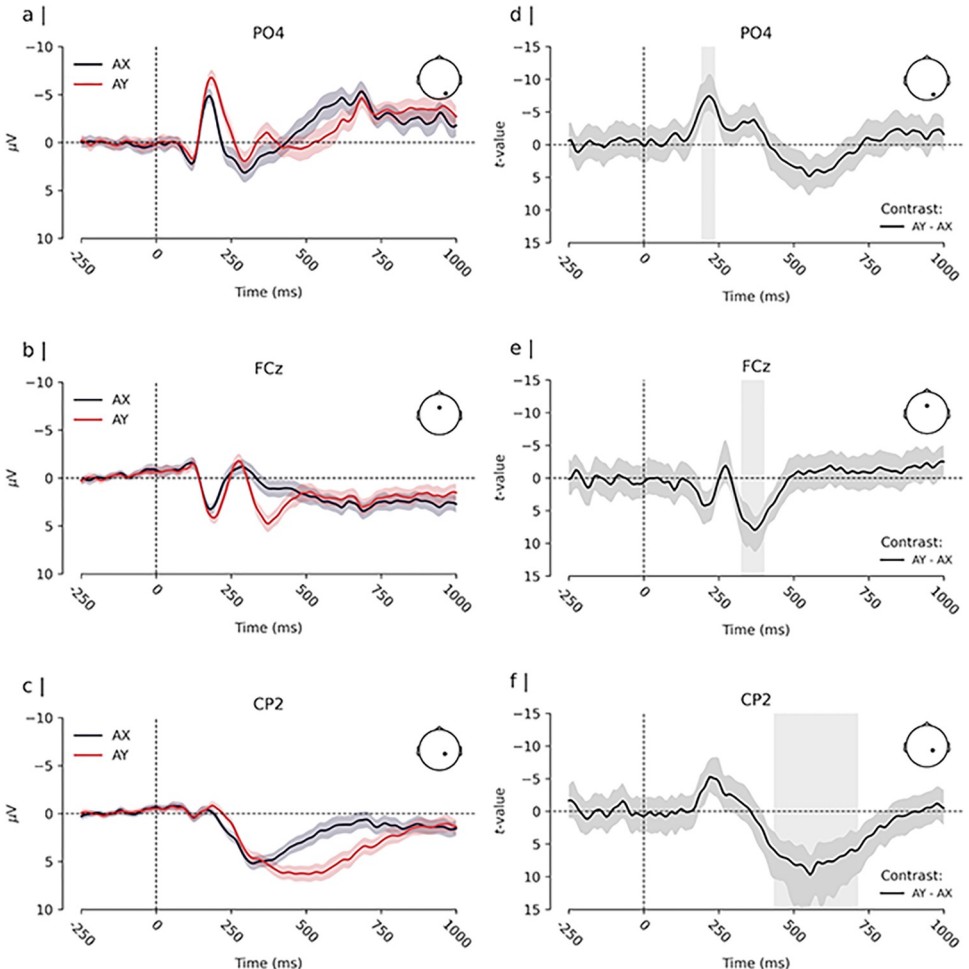

**Fig 6. Probe-evoked ERPs and estimated difference.** *Note.* Panels a), b), and c) depict the Event-Related-Potential (ERP, i.e., the grand average of the participants' mean evoked amplitude response) for "X" and "Y" probes in "A" cue trials (i.e., the average probe evoked activity by "AY" and "AX" pairs, time locked to probe onset). Time 0 ms depicts the onset of the probe stimulus. Depicted are channels that were located within clusters of significant differences between cue "X" and "Y" probes. The solid lines depict the ERP for each condition. The coloured shades surrounding the lines express the uncertainty of the estimates with a 99% within-subjects confidence interval [cf. 78]. Panels d), e), and f) depict the time course of the estimated difference effect at each corresponding channel. The solid line depicts the t-value for the contrast "Y"—"X" probe, and the shaded region surrounding the line expresses its uncertainty with a 99% confidence interval. Vertical grey bars highlight times of significant difference between conditions.

First, we tested the classifiers' ability to decode the identity of "AX" and "BX" trials based on their evoked amplitude patterns. In these trials, the probe was the same, but the required behaviour was different. Results revealed that classifier performance was highest during the cue processing and maintenance interval (i.e., 500–2500 ms, see Fig 7A). As depicted in Fig 7B, diagonal decoding performance, which evaluated the performance of classifiers trained and tested at corresponding time windows, revealed significant decoding of "AX" and "BX" trials as early as 180 ms (AUC = 0.57, 99% CI = [0.54, 0.61]). This performance improved beginning at approximately 300 ms and peaked at approximately 600 ms into the cue mainte-nance interval (AUC = 0.68, 99% CI = [0.63, 0.73]). The decoding remained significantly above chance up to 1100 ms into the cue maintenance interval (AUC = 0.60, 99% CI = [0.56, 0.63]), before returning to chance level at around 1600 ms. Interestingly, significant decoding

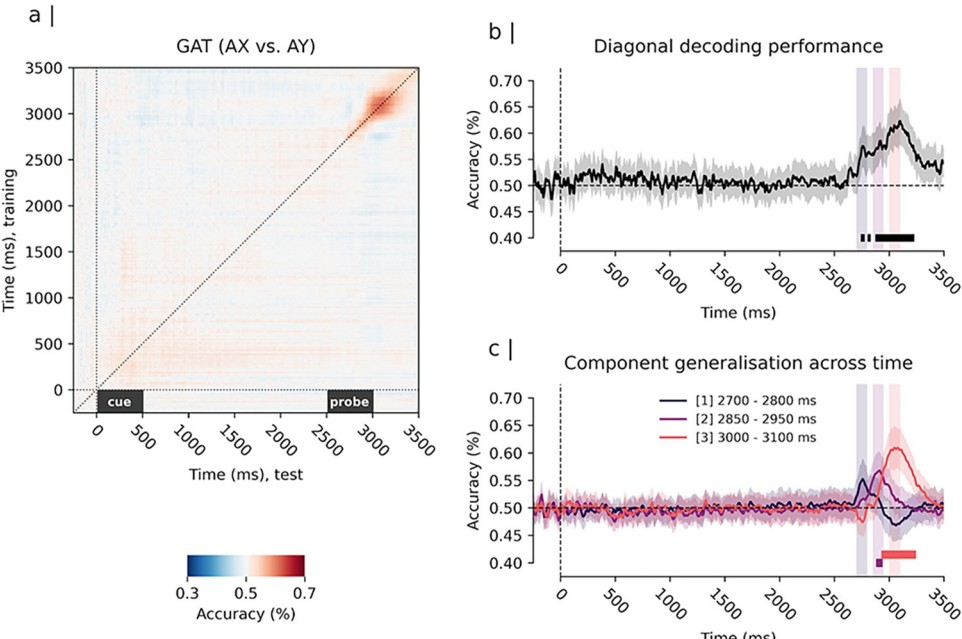

**Fig 7. Temporal generalisation matrix and multivariate pattern analysis result for the contrast "AX" vs "BX" trials.** *Note.* Panel a) depicts the generalisation across time (GAT) matrix for the classifier performance averaged across subjects. Training times are depicted on the y-axis and test times on the x-axis. Time 0 ms depicts the onset of the cue stimulus. Time 2500 ms depicts the onset of the probe stimulus. Panel b) depicts the multivariate pattern decoding performance along the diagonal of the GAT matrix (i.e., diagonal decoding performance). The solid line in b) shows the average diagonal decoding performance and the shaded region surrounding it expresses its uncertainty with a 99% confidence interval. Significant above chance ($> 50\%$) diagonal decoding was observed at approx. 180 ms and between 300 and 1600 ms ($p < 0.01$, Bonferroni corrected). (C) GAT decoding performance for selected time windows of interest matching the peak of the patterns identified by mass univariate analysis: 180–190 ms, 550–650 ms, 1000–1100 ms from cue presentation. The depicted performance time-courses reflect the average of performance time-courses of all classifiers trained in the respective time window. Classifiers trained on the 180–190 ms time window showed no significant above chance decoding after correction for multiple comparisons. Classifiers trained on the 550–650 ms showed significant above change performance from approx. 300–900 ms ($p < 0.01$, Bonferroni corrected). Classifiers trained on the 1000–1100 ms time window showed significant above chance performance between approx. 700 and 1500 ms ($p < 0.01$, Bonferroni corrected); Overlap (i.e., simultaneous significant decoding of both classifiers) was observed between approx. 700 and 900 ms from cue presentation.

of "AX" and "BX" trials was also observed during the probe processing time (see Fig 7B, time > 2500 ms). The peak performance (AUC = 0.58, 99% CI = [0.55, 0.62]) was achieved at approximately 2800 to 2900 ms from post-cue, corresponding to approximately 300 to 400 ms post-probe. These findings mirror the results from the mass univariate analysis approach, with enhanced diagonal performance coinciding with the time windows of the identified patterns of differences between "A" and "B" cues.

GAT analyses revealed significantly different dynamics of the classifiers trained on different cue amplitude response patterns (see Fig 7C). Classifiers trained on early evoked occipital-parietal activity failed to generalise to other stages of the cue maintenance interval. This indicates that the underlying neural generator is temporally specific and is only transiently activated at the beginning of the information processing stream. In contrast, classifiers trained on the cue-evoked parietal positivity and cue-evoked fronto-central negativity were able to generalise to other stages of the cue maintenance interval, sharing a substantial degree of overlap (see Fig 7C, time: 500–1000 ms). This indicates a more sustained activation of the neural generators during the preparatory recruitment of control. The parietal positivity classifier achieved the highest decoding performance of all classifiers and significantly generalised

between approximately 300–900 ms. In contrast, the fronto-central negativity classifier significantly generalised between ~ 700 and 1500 ms.

Second, we tested classifiers' ability to distinguish between "AX" and "AY" trials based on their evoked amplitude patterns. In these trials the cue was the same but the required behaviour was different. Results indicated that classifier performance was highest during the probe processing interval (i.e., 2500–3000 ms, corresponding to 0 to 1000 ms post-probe; see Fig 8A). As depicted in Fig 8B, diagonal decoding performance indicated that "AX" and "AY" pairs could be significantly decoded beginning around 2700 ms post cue (i.e., 200 ms from probe presentation; AUC = 0.57, 99% CI = [0.54, 0.61]). Diagonal decoding performance progressively increased from this time point, reaching its peak at approximately 3100 ms (i.e., 600 ms from probe presentation, AUC = 0.66, 99% CI = [0.62, 0.70]). Thereafter, diagonal decoding performance reverted to chance level towards the end of the analysis time window. These effects mirror those from the mass univariate analysis approach, with peaks in diagonal performance aligning with the time windows of the probe-evoked early parietal-occipital negativity, frontal positivity, and subsequent parietal positivity.

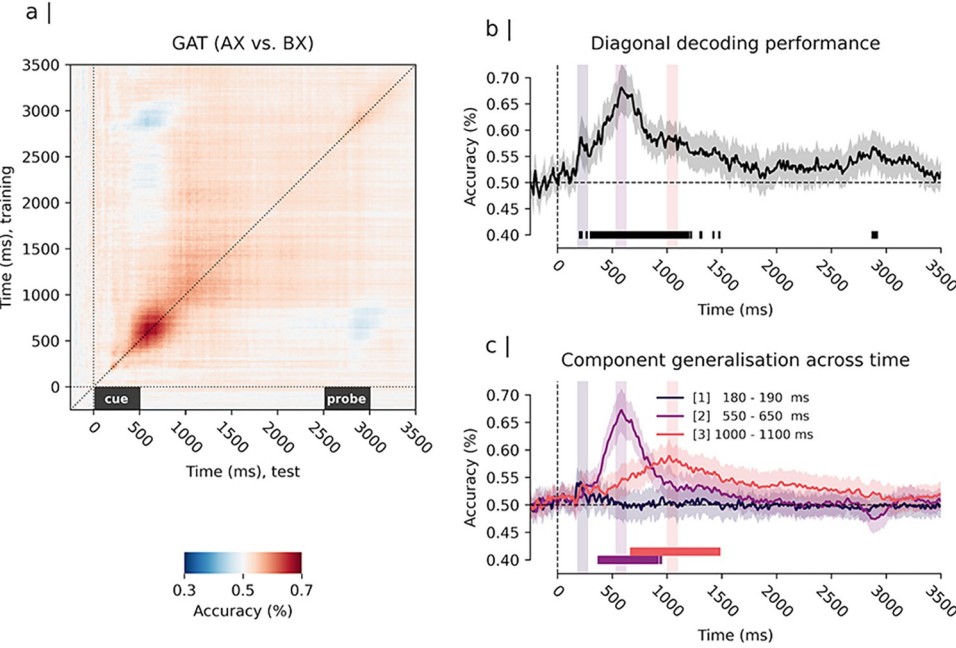

**Fig 8. Temporal generalisation matrix and classifier performance for the contrast "AX" vs "AY" trials.** *Note*. Panel a) depicts the generalisation across time (GAT) matrix for the classifier performance averaged across subjects. Training times are depicted on the y-axis and test times on the x-axis. Time 0 ms depicts the onset of the cue stimulus. Time 2500 ms depicts the onset of the probe stimulus. Panel b) depicts the multivariate pattern decoding performance along the diagonal of the GAT matrix (i.e., diagonal decoding performance). The solid line in b) shows the average diagonal decoding performance for the contrast "AY" vs "AX". The shaded region surrounding it expresses the uncertainty with a 99% confidence interval. Significant above chance (> 50%) diagonal decoding was observed between approx. 2700 ms and 3500 ms from cue presentation (i.e., corresponding to 300 and 900 ms from probe presentation; p < 0.01, Bonferroni corrected). (C) GAT decoding performance for selected time windows of interest matching the peak of the patterns identified by mass univariate analysis: 2700–2800 ms, 2850–2950 ms, 3000–3100 ms post-cue presentation. The depicted performance time-courses reflect the average of performance time-courses of all classifiers trained in the respective time window. Classifiers trained on the 2700–2800 ms time window showed above chance decoding during the same time window but no generalisation across time effects. Classifiers trained on the 2850–2950 ms time window showed significant decoding performance from 2800–3000 ms (p < 0.01, Bonferroni corrected). Classifiers trained on the 3000–3100 ms time window showed significant above chance performance between 2900 and 3300 ms (p < 0.01, Bonferroni corrected); Overlap (i.e., simultaneous significant decoding) was observed between 2800 and 3000 ms post-cue.

GAT analysis revealed significantly different dynamics among the classifiers trained on different probe-evoked amplitude response patterns (see Fig 8C). Similar to the classifiers trained on early cue-evoked activity, classifiers trained on early probe evoked occipital-parietal activity failed to generalise to other stages of the information processing stream. Conversely, classifiers trained on the frontal and parietal positivity patterns significantly generalised for approximately 200 ms, exhibiting a smaller degree of overlap compared to the cue-classifiers (see Fig 8C, time: 3000 ms, i.e., 500 ms post-probe). The frontal positivity classifier significantly generalised between approximately 2800–3000 ms (i.e., 300–500 ms post-probe). Furthermore, a weak but significant generalisation effect of the frontal positivity classifier to cue-evoked activity was found (time: 150–170 ms post-cue; AUC = 0.53, 99% CI = [0.52, 0.55]), suggestive of a functional relationship between early cue evoked activity and frontal positivity responses to the probe. The parietal positivity classifier achieved the best decoding performance of all probe classifiers and significantly generalised between approximately 2900 and 3400 ms (i.e., 400–900 ms post-probe).

**Relationship between behavioural performance and evoked amplitude response.** Finally, we examined whether there was a relationship between behavioural performance in the DPX and the strength of evoked amplitude response patterns. To this end, we computed two signal detection theory-based measures [cf. 79], *d' Context* and the *A-cue response bias*, and used them to predict differential amplitude responses in a simple regression model over the whole scalp. *d' Context* and the *A-cue response bias* measures quantify the extent to which participants depended on cue or probe information to make decisions during the task [80]. Each measure provides a different perspective on this effect, as explained below.

*d' Context* was calculated by comparing the rate of correct responses in "AX" trials to the rate of incorrect responses in "BX" trials ($d'_{context} = Z_{AX\ correct} - Z_{BX\ incorrect}$). It can be seen as a measure of reliance on cue information (i.e., the context) to increase performance during the task (i.e., higher scores mean better performance). For instance, someone who relies strongly on cue information is more likely to prepare a target response in "A"-cue trials and, concurrently, more likely to prepare a non-target response "B"-cue trials. Thus, they would show a high number of correct responses in "AX"-trials and a low number of errors in "BX"-trials. In contrast, someone who is less reliant on cue information could still perform well in "AX" but would be less accurate in "BX" trials, thus leading to a lower score in d' Context [43].

Conversely, the *A-cue response bias* was calculated by comparing the rate of correct responses in AX trials to the rate of incorrect responses in "AY" trials ($A_{bias} = 0.5 \times [Z_{AX\ correct} + -Z_{AY\ incorrect}]$). This measure reflects the degree of behavioural bias associated with "A"-cues. While this bias proves beneficial in "AX" trials, it results in poorer performance in "AY" trials —implying that higher scores correspond to worse performance in "AY" trials.

Our analysis revealed significant moderation effects of *d' Context* on the estimated difference between "B" and "A" cues (see S3 Fig in S1 File). Higher *d' Context* scores were associated with a more positive amplitude response for "B" cues compared to "A" at parietal channels at approximately 300 and 550 ms (peak: PO3 at 562 ms; *b* = 1.27 µV, 99% CI = [0.16, 2.23]). In addition, higher *d' Context* scores were associated with a more pronounced negative amplitude response for "B" cues compared to "A" at frontal channels at approximately 1130 ms post cue (i.e., towards the end of the cue retention interval; peak: channel FC2 at 1133 ms, *b* = -0.98 µV, 99% CI = [-2.43, 0.32]).

Further, we found significant moderation effects of *d' Context* on the estimated difference between "AX" and "AY" probes (see S4 Fig in S1 File). Higher *d' Context* scores were associated with a stronger positivity effect for "AY" compared to "AX" probes at parietal-occipital channels approximately between 400 and 600 ms post probe (peak: channel O1, *b* = 1.68 µV, 99% CI = [-0.19, 3.60]). During the same time period, higher d' Context scores led to more

negative amplitude responses for "AY" compared to "AX" probes at frontal channels (peak: channel Fz, *b* = -1.98 µV, 99% CI = [-3.79, -0.16]).

Finally, we found very few significant effects of the *A-cue response bias* on the estimated difference between "B"- and "A"- cues (see S5 Fig in S1 File) and the estimated difference between "AY"- and "AX" probes (see S6 Fig in S1 File). Results indicated that higher *A-cue response bias* scores were mainly associated with more negative amplitudes responses for "AY" compared to "AX" probes at central channel at approximately 100 and 300 ms post probe (peak: Channel CP1, *b* = -0.59 µV, 99% CI = [-1.15, -0.03]).

## Discussion

Cognitive control is thought to rely on a neural network spanning posterior parietal and frontal brain regions [6]. Preparatory and adaptive control are believed to serve complementary purposes, enabling flexible goal-directed behaviour in dynamic environments by recruiting neural resources at different stages of decision-making [46]. To further elucidate the temporal dynamics of preparatory and adaptive control recruitment, our study employed a combination of hierarchical mass univariate and MVPA procedures. We examined EEG amplitude responses evoked during the performance of the DPX–a task frequently used to measure the contributions of preparatory and adaptive cognitive control to behavioural performance. Our goal was to provide a more comprehensive assessment of the EEG amplitude patterns associated with the preparatory and adaptive recruitment of control, as well as to delineate their functional timing and spatio-temporal evolution throughout the information processing stream.

### Early evoked occipital-parietal response patterns

Analysis of the cue-evoked amplitude responses revealed significant differences between cues that encouraged preparatory recruitment of control (i.e., "B" cues or highly preparatory cues) and more ambiguous ones (i.e., "A" cues). Notably, around 170–200 ms post-cue presentation, highly preparatory cues exhibited an enhanced negative amplitude response at parietal-occipital channels (e.g., P5, PO7, P6, PO8) and an enhanced positive amplitude response at frontal channels (e.g., FC1, F4). Furthermore, our analyses revealed similar early amplitude response patterns for probes that appeared subsequently. Probes, which signalled an unexpected deviation from the prepared behavioural response (i.e., "Y" probes after an "A" cue), elicited a more negative amplitude response pattern at parietal-occipital channels and a more positive amplitude response at frontal channels compared to expectation-confirming probes (i.e., "X" probes after an "A" cue).

These results are in line with previous research [21, 22]. Past studies focusing on ERP components have shown enhanced occipital N1 amplitudes for highly preparatory cues [21]. Meanwhile, other research has highlighted an enhanced fronto-central positive polarisation of the ERP in the context of these cues [22]. Our findings suggest that highly preparatory cues evoked enhanced amplitude response patterns at both occipital and fronto-central electrodes. This indicates that earlier, seemingly isolated amplitude response effects could be two views on the same experimental effect.

Early evoked EEG amplitude responses have been associated with initial sensory processing [47], benefits from stimulus feature selection [81, 82], and adjustments in neural circuitry before attention is deployed [83]. Although typically linked with specialised perception processes, such as facial feature identification [84], there is growing evidence suggesting that these amplitude response patterns may represent a domain general attentional allocation mechanism [82]. Early occipital and fronto-central amplitude responses have been documented across various stimuli types, from words to figures [85] and spatial navigation cues [86]. Their

association with salience detection in different domains [87] suggests the involvement of consistent neural generators across tasks.

Our results resonate with this viewpoint. The early amplitude response patterns seen in the DPX task could signify a core sensory processing mechanism, paving the way for more robust recruitment of control in subsequent processing stages. Notably, our data indicate that both highly preparatory cues and probes indicating a need for behavioural adjustment amplify the engagement of these mechanisms. In trials with cues that encourage preparatory recruitment of control, there seems to be a heightened recruitment of attentional resources, potentially facilitating a subsequent activation of the needed "task-set" [cf. 24]. Similarly, increased attentional recruitment by deviant probes seems essential to prompt the system for behavioural adaptation.

Time generalisation analysis indicated that these early spatio-temporal patterns were highly specific in their ability to discriminate between different types of cues and probes. Namely, classifiers trained on this early amplitude response could decode the identity of the stimuli during the same time window, but their performance did not generalise to other stages of the information processing stream (see **Fig 7**, for instance). This suggests that their neural generator facilitated attentional allocation to features of the stimulus but contributed less to subsequent processing of stimulus information (e.g., behavioural configuration processes). Indeed, it has been proposed that early negative parietal amplitude responses may be related to more pre-attentive adjustments in the underlying neural circuitry than behavioural preparation or response selection processes [83].

## Mid-range parietal and frontal-parietal positivity patterns

Cues that encourage preparatory recruitment of control elicited a stronger and more sustained positive amplitude response pattern across multiple parietal electrodes than more ambiguous cues. This finding aligns with prior research. Studies examining ERP components have reported enhanced positive amplitude responses for preparatory cues, particularly at parietal-occipital electrodes [22]. Moreover, similar amplitude response patterns have been identified in task switching paradigms [24, 88]. In these paradigms, cues that indicate an upcoming change in behavioural requirements generally elicit a more prominent positive polarisation of the ERP in central-parietal regions compared to control cues [88]. Our data builds on these findings, indicating that stimuli which foster proactive recruitment of control induce changes in EEG activity across a wide parietal cluster, with the temporal extent and magnitude scaling with the predictive value of the stimuli.

Central-parietal amplitude responses have been related to the stimulus evaluation stage where stimuli are classified into categories, such as targets or distractors, and "go" or "no-go" commands, allowing for a selection of appropriate responses to the environment [23, 35, 89, 90]. Supporting this notion, research has indicated that the magnitude of these responses correlates with performance benefits [24]. In our study, a clear relationship emerged between behavioural performance in the DPX and the differential parietal amplitude response observed for highly preparatory versus ambiguous cues. Specifically, participants with higher d' Context scores exhibited a more pronounced differential positivity at central and fronto-central channels around 600–800 ms post-cue. Participants' *d' Context* scores reflect their reliance on cue information in the DPX. Hence, a more positive amplitude response for cues than encouraging behavioural preparation could indicate increased engagement of frontal brain regions [91, 92] supporting the configuration of the underlying neurocognitive architecture in accordance with the demands at hand.

Central-parietal amplitude responses are posited to denote an ongoing stimulus evaluation stage [23, 89, 90]. In this stage, the system classifies stimuli into categories such as targets or

distractors, and "go" or "no-go" imperatives, enabling the selection of appropriate responses [23, cf. 35]. Previous research has shown that the magnitude of these response patterns aligns with condition-specific performance benefits [24]. Consistent with this, our study revealed a relationship between behavioural performance in the DPX and the sustained central-parietal amplitude response. Subjects with higher *d' Context* scores displayed an additionally enhanced differential positivity effect for preparatory cues at fronto-central electrodes. An individual's *d' Context* score can be understood as their reliance on cue information during the DPX. Thus, stronger fronto-central amplitude responses for these preparatory cues may indicate increased engagement of frontal brain structures [91, 92], supporting the configuration of the neural circuitry that facilitates the response selection process.

Similarly, probes that indicated an unexpected deviation from the anticipated behavioural response elicited an enhanced central-parietal positivity, when compared to expectation-confirming probes. Preceding this parietal positivity, we also observed an enhanced frontal positivity for these probes. We interpret this sequential enhancement of frontal and parietal positivity patterns as the cascading activation of neurocognitive processes intended to overcome behavioural biases. The fronto-central positivity could represent the recruitment of frontal attentional resources. This recruitment is triggered when a stimulus, contrary to expectations, prompts the system to revise its current action strategy [91]. The subsequent heightened parietal positivity pattern could reflect a heightened deployment of cognitive resources needed for adequate stimulus-response mapping. This notion is supported by our finding that subjects who were more reliant on preparatory processes (as indicated by a higher *d' Contexts* scores), also showed a more pronounced differential parietal positivity effect when probes signalled the need for late behavioural adaptation.

Time generalisation analysis revealed that classifiers trained on this parietal-positivity response achieved the best decoding performance among all classifiers. In addition, these classifiers were able to generalise over large proportions of the cue and probe processing intervals. This indicates that the underlying patterns of brain activity were sustained in a stable form for several, partially overlapping, tens of milliseconds. Furthermore, parietal positivity classifiers showed a substantial degree of overlap with the subsequent fronto-central negativity pattern evoked by cue-stimuli and the fronto-central positivity evoked by probe stimuli. These results suggest a rapidly succeeding and partially overlapping activation of underlying brain structures, each for a short time period, in response to behaviourally relevant stimuli.

## Late fronto-central negativity patterns

Finally, our analysis revealed an enhanced frontal-central negativity for cues that encouraged preparatory recruitment of control compared to their more ambiguous counterparts. This effect peaked at approximately 1400 ms post-cue, consistent with prior research. Enhanced fronto-central ERPs, such as the CNV [17], have been identified for cues that signal the preparation of a behavioural response in comparable tasks [21]. These amplitude response patterns have been linked to anticipatory attention [18] and adjustments in response caution [20] ahead of goal-relevant events. In our study, the pronounced fronto-central negativity for these highly preparatory cues could reflect the engagement of an anticipatory process that allows the system to bias response execution, which in turn facilitates quicker reactions during these trials.

## Limitations and implications for further research

Our study extends previous research by demonstrating that the recruitment of cognitive control affects amplitude responses across multiple scalp areas. Instead of merely influencing

individual ERPs, the recruitment of control appears to initiate a cascade of amplitude fluctuations in the scalp-recorded EEG, a finding that we firmly believe better represents the processing of information in the underlying neural circuitry. This is a key finding because it provides several advantages in terms of construct and ecological validity.

Traditional ERP analyses often rely on averaged responses to specific stimuli, potentially overlooking the dynamic nature of cognitive processes as they occur in real-world settings [27]. In contrast, our study demonstrated that using mass-univariate and multivariate analysis methods, we can capture the spatio-temporal evolution of EEG responses more comprehensively [33]. This allows for a more nuanced understanding of how cognitive control is recruited and implemented over time [36]. Our findings indicate that cognitive control is not a static process localized to specific brain regions but rather arises from a dynamic interplay of neural structures distributed across the brain [6]. This is in line with the concept of distributed neural processing [10, 93], where cognitive functions are supported by networks of interacting regions rather than isolated areas, highlighting the importance of temporal dynamics in understanding cognitive control [10, 94].

Furthermore, our results have practical implications for the assessment and training of cognitive control. By providing a more detailed mapping of EEG responses, we believe that our approach could enhance the precision of brain modulatory interventions (e.g., neurofeedback, transcranial direct current stimulation) aimed at improving brain function [95, 96]. Understanding that cognitive control is a cascade of processes rather than a static function can lead to more targeted treatments. For example, in disorders like ADHD [97, 98] or schizophrenia [99], where selective deficits in cognitive control are prevalent, interventions can be designed to specifically target the dynamic interplay of neural circuits. This could involve tailored neurofeedback protocols [cf. 100] that focus on enhancing the temporal coordination of these processes, potentially leading to more effective treatments and better outcomes for patients.

One further limitation of our study is that both target and non-target responses were performed using the same (dominant) hand. Originally, this decision was made to minimize influences introduced by expertise using the non-dominant hand for either response. At the same time, this could have introduced an unwanted lateralisation effect during neural processing (more negative amplitude responses over the right occipital lobe for unpredicted probes). However, the lateralisation effects present in our data seemed to be more indicative of sensory processes (e.g., early attention allocation to the probe) rather than motor preparation. Nevertheless, more specialized studies are needed to rule out contaminants introduced by the experimental setting. Future research should consider using responses from both hands to determine if this influences the observed brain activity patterns. This approach could help clarify the impact of motor responses on lateralization and provide a more comprehensive understanding of the neural mechanisms underlying cognitive control.

Our current experimental design only allows us to make informed hypotheses regarding the specific sub-processes activated by the recruitment of cognitive control. Also, we cannot satisfactorily rule out the potential influence of confounds introduced by characteristics of the DPX. For instance, an alternative interpretation for our findings could be that subjects were simply more accustomed to seeing certain types of cues and probes (e.g., "A"-cues and "X"-probes). Thus, stimulus frequency could influence the amplitude response patterns recorded during the task. Indeed, stimulus frequency has been shown to influence the amplitude of early evoked parietal-occipital responses, such as the N170 [101], with amplitude decreasing as stimulus frequency increases. Stimulus frequency could therefore explain reduced early amplitude responses to more frequent cues, which, in our study, also predicted subsequent behavioural demands less accurately. However, we observed that amplitude modulations were not specifically tied to the infrequent stimuli themselves (e.g., "Y" probes). Instead, strong

amplitude modulations were elicited when stimuli were behaviourally relevant (e.g., "Y" probes in "A"-cue trials). In contrast, there was no similar effect when infrequent stimuli had no impact on behaviour (e.g. "Y" probes in "B"-cue trials) even though the stimulus itself was generally less frequent. Therefore, it seems unlikely that stimulus frequency alone can fully account for the differences between conditions. Nonetheless, future studies should consider controlling for or isolating stimulus frequency effects to provide a clearer understanding of the processes at play. Future research should further explore the specific features of cues and probes that drive these enhanced parietal and fronto-central responses, as well as investigate how these neural differences translate to behavioural performance in tasks requiring attention, evidence accumulation, and decision-making.

Lastly, there's a growing consensus in the literature indicating that both preparatory (often termed proactive) and adaptive (or reactive) recruitment of control offer distinct advantages and limitations [46]. While preparatory control can be more effective by enabling a focus on goal-relevant behaviour and recruiting control before the occurrence of relevant events [102], its efficiency hinges greatly on the validity and reliability of the contextual cues prompting preparation, as these cues inform predictions about likely scenarios [6, 8]. Conversely, adaptive control provides flexibility in responding to unexpected shifts, conflicts, or errors [103]. Yet, it often results in slower reactions compared to the anticipatory benefits of proactive control [104]. Existing research points to a competition for shared neurocognitive resources between these two control strategies [105]. Moreover, behavioural outcomes seem to stem from a balance between the two [12]. The intricate dynamics between these mechanisms, their functional timing, and the neural architecture supporting them remain subjects of intensive study [1, 10, 106]. Our findings, in this context, offer promising insights into the temporal interplay of these control systems. By capturing the cascade of amplitude fluctuations in scalp-recorded EEG associated with the recruitment of control, we provide an integrated account of how these systems may be functioning in concert. The timing of these fluctuations, particularly their onset and overlap, sheds light on the potential sequence and interactions of preparatory and adaptive controls at a neural level. As we move forward, a key challenge lies in examining the universality of our findings across varied populations and experimental setups. Furthermore, probing the role of individual differences in shaping these neural responses becomes crucial. It appears crucial to probe the role of individual differences in shaping these neural responses. Specifically, factors like motivation [107], the influence of incentives [108], working memory capacity [80, 109], and overall cognitive ability might differentially modulate the interplay between preparatory and adaptive controls, thereby potentially altering the observed neural patterns and their relation to behavioural outcomes.

## Conclusion

In this study, we demonstrate that the evoked amplitude response to cues and probes corresponded to an extended sequence of distinct and partially overlapping brain activation patterns, as opposed to single neural signatures for preparatory and adaptive recruitment of control. Importantly, our findings are in line with previous research reporting differential amplitude responses for behaviourally relevant cues and probes. Our results provide a direct replication and extension of these findings by demonstrating that the differential amplitude response cue and probe stimuli in the DPX is not constrained to individual channels and time-windows. In contrast, our results firmly demonstrate that previously reported and seemingly isolated differences between conditions can in fact be integrated into a sequence of distinct spatio-temporal amplitude response patterns, indicating a cascadic propagation of brain activity between sensory processing and anticipatory behavioural configuration. These effects

complement the results of the behavioural analysis by showing that differences in behavioural performance are tracked by significant differences in the processing of the cues and probes. Furthermore, our multivariate spatio-temporal approach provides additional information about the functional timing and organisation of the candidate neurocognitive processes underlying the engagement in preparatory and adaptive cognitive control. These results provide a more representative approximation and mapping of the neurocognitive mechanisms that enable flexible behavioural adaptation in context dependent decision-making [cf. 12, 110] compared to more traditional ERP methods [29, 33].

## Supporting information

**S1 File. Supporting information.** Supplemental tables and figures for the results described in the main manuscript.
(DOCX)

## Author Contributions

**Conceptualization:** José C. García Alanis, Mira-Lynn Chavanon, Martin Peper.

**Formal analysis:** José C. García Alanis.

**Investigation:** José C. García Alanis, Malte R. Güth.

**Methodology:** José C. García Alanis, Malte R. Güth.

**Software:** José C. García Alanis.

**Visualization:** José C. García Alanis.

**Writing – original draft:** José C. García Alanis.

**Writing – review & editing:** José C. García Alanis, Malte R. Güth, Mira-Lynn Chavanon, Martin Peper.

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
