## [Decision Letter · Decision Letter 0]

9 Feb 2024

PONE-D-23-34766Neurocognitive Dynamics of Preparatory and Adaptive Cognitive Control: Insights from Mass-Univariate and Multivariate Pattern Analysis of EEG dataPLOS ONE

Dear Dr. García Alanis,

Thank you for submitting your manuscript to PLOS ONE. After careful consideration, we feel that it has merit but does not fully meet PLOS ONE’s publication criteria as it currently stands. Therefore, we invite you to submit a revised version of the manuscript that addresses the points raised during the review process.

We look forward to receiving your revised manuscript.

Kind regards,

Yansong Li

Academic Editor

PLOS ONE

Journal Requirements:

4. Thank you for stating the following in the Acknowledgments Section of your manuscript: "Research was supported in part by grants awarded to José C. García Alanis by the German Academic Exchange Service (DAAD)."

Please remove any funding-related text from the manuscript and let us know how you would like to update your Funding Statement. Currently, your Funding Statement reads as follows:"Research was supported in part by grants awarded to José C. García Alanis by the German Academic Exchange Service (DAAD)."

Additional Editor Comments (if provided):

Reviewers' comments:

Reviewer's Responses to Questions

**Comments to the Author**

1. Is the manuscript technically sound, and do the data support the conclusions?

Reviewer #1: Yes

Reviewer #2: Yes

2. Has the statistical analysis been performed appropriately and rigorously? 

Reviewer #1: Yes

Reviewer #2: Yes

3. Have the authors made all data underlying the findings in their manuscript fully available?

Reviewer #1: No

Reviewer #2: Yes

4. Is the manuscript presented in an intelligible fashion and written in standard English?

Reviewer #1: Yes

Reviewer #2: Yes

5. Review Comments to the Author

Reviewer #1: The reviewed paper reports interesting results from an investigation into dynamic interplay between preparatory and adaptive cognitive control systems. The findings show distinct types of control processes engaging differing, but partially overlapping brain activation patterns. Furthermore, performance was associated significantly with centro-parietal and fronto-central brain activations.

After reading the manuscript, I can confirm that the study is within the scope of PloS One, meets its scientific requirements and will be an important addition to the current literature.

My comments concern only: 1. clearer description of the conditions/results to make this manuscript more accessible for broader readership, and 2. final interpretation of findings in terms of a direct application (e.g., in the context of clinical populations). 1. For instance, DPX conditions could be defined in plain language already in the Method section (e.g., Cue A as ambiguous, Cue B as highly preparatory stimulus). This is explained in the Discussion, but having this information earlier (and re-iterated) would decrease demand on working memory for those who do not work with this particular paradigm. Similarly, labels for measures discussed in the Results section can be replaced by the studied effect (e.g., Stronger reliance on cues was associated with more pronounced brain response following highly preparatory cues…).

I noticed some effects were lateralised. Could the authors comment on that?

In short, advantages of the approach taken by the authors in terms of ecological validity is clear, but the Results and Discussion could sell this point better if the outcomes would be discussed in a more concrete manner and touching on implications.

Minor comments:

Information about participants’ eyesight and hand dominance should be included in the Methods.

The paradigm description could include more specific information – e.g., target response = pressing green (left) button. Were both target and non-target responses performed by the same (dominant) hand? If so, could this have influenced brain response patterns in any way?

All results should be reported in one metric (msec vs sec).

Line 506 – I am not sure what “more pronounced positive negative response” means.

Lines 817-820 - I do not understand the reasoning here, please reword.

Figures – indicating presentation of a probe stimulus would facilitate understanding of the results (the same approach is applied in the text).

Table S2 – Are p values correct for the gender covariate? Also, confidence interval values seem to be crossing zero.

Reviewer #2: In their current study, García Alanis et al. apply advanced EEG methods with the goal to separate neurocognitive mechanisms of preparatory versus adaptive cognitive control. The authors investigate an interesting research question. The methods are very sophisticated, the quality of the figures is extremely high, and the support of open science practices (especially open data) is a further plus. Nevertheless, I have some comments that might be worth to consider before publication. I am convinced that the manuscript will be of high interest to the readers of PLOS ONE.

1. The authors discuss the relevance of the P3b component and the central parietal positivity (e.g., page 4). I have repeatedly asked myself to what extent these components - especially the second one - are related to modulations of the late positive potential. Robust modulations of the late positive potential have often been shown with other cue-related designs, e.g., fear conditioning (for examples, see Bacigalupo & Luck, 2018, Psychophysiology, https://doi.org/10.1111/psyp.13015; Sperl et al., 2021, NeuroImage, https://doi.org/10.1016/j.neuroimage.2020.117569). If there are any links to the late positive potential literature, it might be worth mentioning them explicitly, as this will increase readership. I could imagine other readers asking themselves the same question as me.

2. The authors use a significance threshold of p < 0.01. This is a plus, because threshold is smaller than in most neuroscientific studies, supporting the robustness of the findings. Nevertheless, I wonder whether there was a particular reason for choosing such a low threshold - for example, are the methods used particularly susceptible to false-positive findings?

3. The authors write that all participants reported no current use of prescription drugs. Please clarify: Did this also include oral contraceptives (given the known role of estradiol levels on findings from learning experiments).

4. As I understand it, the EEG artifact screening procedure was completely automated, right? I am a big fan of automated procedures. Nevertheless, I am always somewhat skeptical when a (final) visual check of the data is completely missing, because (systematic) artifacts in the data may not be detected or may be excluded even though they could possibly be corrected. That's why I ask myself: Was there also a final visual check of the data?

5. The authors' methodological approach is impressive and attempts to address the limitations of previous studies. At the same time, however, the methodological approach is also extremely complex, and there is a risk that readers will drop out in between. It might therefore be helpful to create a schematic diagram that illustrates the rationale of the methodological approach in the form of arrow diagrams, etc. This could help readers to understand the procedure faster and better.

6. Page 18, line 432: “Th earliest …” – an “e” is missing after the “Th”.

7. On page 25, the authors write “Each measure provides a different perspective on this effect.” At first, when I read this sentence, I thought there would be no further explanation of which different perspectives were meant. Then, as I read on, I realized that the authors do actually provide an explanation. It might be worth to improve the reading flow here, e.g.: “Each measure provides a different perspective on this effect, as explained below.”

6. PLOS authors have the option to publish the peer review history of their article (what does this mean?). If published, this will include your full peer review and any attached files.

Reviewer #1: No

Reviewer #2: **Yes: **Matthias F.J. Sperl

---

## [Author Response · Author response to Decision Letter 0]

9 Aug 2024

Dear Yansong Li (academic editor, PLOS ONE),

We would like to thank you and the reviewers for the helpful comments and suggestions on our manuscript. We kindly hope that you and the reviewers can excuse the major delay in our revision. Below, we have provided our responses to each of the reviewers' comments. Our responses are indented and appear directly below each comment.

José C. García Alanis

(on behalf of the authors)

Comments to the author:

Reviewer #1:

The reviewed paper reports interesting results from an investigation into dynamic interplay between preparatory and adaptive cognitive control systems. The findings show distinct types of control processes engaging differing, but partially overlapping brain activation patterns. Furthermore, performance was associated significantly with centro-parietal and fronto-central brain activations.

After reading the manuscript, I can confirm that the study is within the scope of PloS One, meets its scientific requirements and will be an important addition to the current literature.

• Response:

We thank Reviewer #1 for their positive feedback and for acknowledging the significance of our study. We appreciate your kind words and are glad that our findings are considered a valuable contribution to the current literature.

My comments concern only: 

1. clearer description of the conditions/results to make this manuscript more accessible for broader readership, and 2. final interpretation of findings in terms of a direct application (e.g., in the context of clinical populations). 

• Response:

We thank Reviewer #1 for their valuable comments and suggestions. We have revised the manuscript accordingly, aiming to make it more accessible for a broader readership. Additionally, we have extended the discussion to include an interpretation of the findings in terms of their direct application in the context of clinical populations.

For instance, DPX conditions could be defined in plain language already in the Method section (e.g., Cue A as ambiguous, Cue B as highly preparatory stimulus). This is explained in the Discussion, but having this information earlier (and re-iterated) would decrease demand on working memory for those who do not work with this particular paradigm. Similarly, labels for measures discussed in the Results section can be replaced by the studied effect (e.g., Stronger reliance on cues was associated with more pronounced brain response following highly preparatory cues…).

• Response:

Thank you for this valuable suggestion. We have revised the Methods section to define the DPX conditions in plain language, providing clearer descriptions of the expected effects for each condition and the facets of cognitive control they operationalize. Additionally, we have added descriptive labels to the cues and probes in the Results section to highlight the studied effects. However, we decided not to replace the more neutral style in the description to avoid making unsupported and circular conclusions. We have provided a thorough and critical interpretation of the results in the Discussion, and we hope these changes make the manuscript more accessible to readers who may not be familiar with the DPX paradigm.

Excerpt from the methods section: (line 220)

”This manipulation was introduced to induce a strong expectation that an “X” probe following an “A” cues, making the “AX” pair the most common and expected condition). Conversely, the “AY”, “BX”, and “BY” pairs each accounted for roughly 10% of the trials and were therefore significantly less frequent. “AY” trials involved a predictive “A” cue followed by an unexpected “Y” probe, requiring participants to override the strong expectation and perform a non-target response – this is a hallmark of reactive cognitive control. In contrast, “BX” and “BY” trials involved an unambiguous and highly preparatory “B” cue, as this cue provided all the information needed for the selection of the correct response, highlighting proactive control.”

I noticed some effects were lateralised. Could the authors comment on that?

• Response:

Thank you for this comment. We now discuss how the experimental setting could have influenced lateralisation effects in the limitations section (see below).

In short, advantages of the approach taken by the authors in terms of ecological validity is clear, but the Results and Discussion could sell this point better if the outcomes would be discussed in a more concrete manner and touching on implications.

• Response:

Thank you for this valuable suggestion. We have extended the discussion section to better highlight the advantages in terms of construct and ecological validity of our approach and to discuss the outcomes in a more concrete manner, touching on their broader implications.

The extend discussion now reads: (line 823)

 “Our study extends previous research by demonstrating that the recruitment of cognitive control affects amplitude responses across multiple scalp areas. Instead of merely influencing individual ERPs, the recruitment of control appears to initiate a cascade of amplitude fluctuations in the scalp-recorded EEG, a finding that we firmly believe better represents the processing of information in the underlying neural circuitry. This is a key finding because it provides several advantages in terms of construct and ecological validity.

Traditional ERP analyses often rely on averaged responses to specific stimuli, potentially overlooking the dynamic nature of cognitive processes as they occur in real-world settings (26). In contrast, our study demonstrated that using mass-univariate and multivariate analysis methods, we can capture the spatio-temporal evolution of EEG responses more comprehensively (32). This allows for a more nuanced understanding of how cognitive control is recruited and implemented over time (35). Our findings indicate that cognitive control is not a static process localized to specific brain regions but rather arises from a dynamic interplay of neural structures distributed across the brain (6). This is in line with the concept of distributed neural processing (10,92), where cognitive functions are supported by networks of interacting regions rather than isolated areas, highlighting the importance of temporal dynamics in understanding cognitive control (10,93).

Furthermore, our results have practical implications for the assessment and training of cognitive control. By providing a more detailed mapping of EEG responses, we believe that our approach could enhance the precision of brain modulatory interventions (e.g., neurofeedback, transcranial direct current stimulation) aimed at improving brain function (94,95). Understanding that cognitive control is a cascade of processes rather than a static function can lead to more targeted treatments. For example, in disorders like ADHD (96,97) or schizophrenia (98), where selective deficits in cognitive control are prevalent, interventions can be designed to specifically target the dynamic interplay of neural circuits. This could involve tailored neurofeedback protocols (cf. 99) that focus on enhancing the temporal coordination of these processes, potentially leading to more effective treatments and better outcomes for patients.

Minor comments:

Information about participants’ eyesight and hand dominance should be included in the Methods.

• Response:

We thank the reviewer for this comment. We have revised the manuscript accordingly. 

The updated Methods section now includes the following information: (line 172)

"All participants had normal or corrected-to-normal vision, assessed using a standardised Landolt C Test, and reported right-handed dominance.

The paradigm description could include more specific information – e.g., target response = pressing green (left) button. Were both target and non-target responses performed by the same (dominant) hand? If so, could this have influenced brain response patterns in any way?

• Response:

Thank you for this comment. We have specified the experimental procedure in the Methods section.

The section now includes the following sentence: (line 186)

“Participants responded by pressing a button with either the index (target) or middle finger (non-target) of the right hand on a Cedrus RB-840 response pad (Cedrus Corporation, San Pedro, CA).”

Also, we have extended the discussion: (line 849)

“One further limitation of our study is that both target and non-target responses were performed using the same (dominant) hand. Originally, this decision was made to minimize influences introduced by expertise using the non-dominant hand for either response. At the same time, this could have introduced an unwanted lateralisation effect during neural processing (more negative amplitude responses over the right occipital lobe for unpredicted probes). However, the lateralisation effects present in our data seemed to be more indicative of sensory processes (e.g., early attention allocation to the probe) rather than motor preparation. Nevertheless, more specialized studies are needed to rule out contaminants introduced by the experimental setting. Future research should consider using responses from both hands to determine if this influences the observed brain activity patterns. This approach could help clarify the impact of motor responses on lateralization and provide a more comprehensive understanding of the neural mechanisms underlying cognitive control.”

All results should be reported in one metric (msec vs sec).

• Response:

Thank you for pointing this out. In an earlier version of the manuscript, we used seconds and missed a couple of instances during the correction to milliseconds. We have now corrected the remaining occurrences, and all results are consistently provided in milliseconds.

These include the figure caption of Figure 2, which now reads:

"Estimated RT is provided in milliseconds (ms) and estimated error rates are provided as proportions (scale 0.0 to 1.0)."

Line 506 – I am not sure what “more pronounced positive negative response” means.

• Response:

Thank you for this comment. This is indeed a typo and has been corrected.

The sentence now reads: (line 526)

“Y” probes also elicited a more pronounced positive negative response at left occipital channels (peak at Iz at approximately 398 ms) compared to “X” probes during the same time window (t(51) = −7.45, d = −1.04, 99% CId [−1.49, −0.59]).

Lines 817-820 - I do not understand the reasoning here, please reword.

• Response:

Thank you. We have rewritten the section.

It now reads: (line 862)

“For instance, an alternative interpretation for our findings could be that subjects were simply more accustomed to seeing certain types of cues and probes (e.g., “A”-cues and “X”-probes). Thus, stimulus frequency could influence the amplitude response patterns recorded during the task. Indeed, stimulus frequency has been shown to influence the amplitude of early evoked parietal-occipital responses, such as the N170 (100), with amplitude decreasing as stimulus frequency increases. Stimulus frequency could therefore explain reduced early amplitude responses to more frequent cues, which, in our study, also predicted subsequent behavioural demands less accurately. However, we observed that amplitude modulations were not specifically tied to the infrequent stimuli themselves (e.g., “Y” probes). Instead, strong amplitude modulations were elicited when stimuli were behaviourally relevant (e.g., “Y” probes in “A”-cue trials). In contrast, there was no similar effect when infrequent stimuli had no impact on behaviour (e.g. “Y” probes in “B”-cue trials) even though the stimulus itself was generally less frequent. Therefore, it seems unlikely that stimulus frequency alone can fully account for the differences between conditions.”

Figures – indicating presentation of a probe stimulus would facilitate understanding of the results (the same approach is applied in the text).

• Response:

Thank you for this suggestion. We intentionally left this information out of the graphical part of the figures to avoid clutter. However, we have now added the following information more explicitly in EEG figure captions to improve readability and understanding:

“Time 0 ms depicts the onset of the <cue/probe> stimulus.”

Table S2 – Are p values correct for the gender covariate? Also, confidence interval values seem to be crossing zero.

• Response:

Thank you for pointing this out. We have corrected the incorrect values, which arose from numerical imprecisions due to the way we coded the analysis and extracted the results for reporting. Additionally, there was a typo in the figure caption that might have led to confusion, and this has also been corrected.

Reviewer #2:

In their current study, García Alanis et al. apply advanced EEG methods with the goal to separate neurocognitive mechanisms of preparatory versus adaptive cognitive control. The authors investigate an interesting research question. The methods are very sophisticated, the quality of the figures is extremely high, and the support of open science practices (especially open data) is a further plus. Nevertheless, I have some comments that might be worth to consider before publication. I am convinced that the manuscript will be of high interest to the readers of PLOS ONE.

• Response:

We thank Reviewer #2 for the feedback and kind words. We have carefully considered and integrated their comments in the revision. We truly believe they have helped improve the quality of the manuscript.

1. The authors discuss the relevance of the P3b component and the central parietal positivity (e.g., page 4). I have repeatedly asked myself to what extent these components - especially the second one - are related to modulations of the late positive potential. Robust modulations of the late positive potential have often been shown with other cue-related designs, e.g., fear conditioning (for examples, see Bacigalupo & Luck, 2018, Psychophysiology, https://doi.org/10.1111/psyp.13015; Sperl et al., 2021, NeuroImage, https://doi.org/10.1016/j.neuroimage.2020.117569). If there are any links to the late positive potential literature, it might be worth mentioning them explicitly, as this will increase readership. I could imagine other readers asking themselves the same question as me.

• Response:

Thank you for this insightful comment. We have integrated your suggestion by mentioning the connection to the late positive potential (LPP) literature.

Excerpt from the introduction: (line 80)

“Additionally, similar late positive potential (LPP) modulations have been observed in other cue-related paradigms, such as fear conditioning (25). These studies further support the idea that parietal positive amplitude responses play a crucial role in processing cues and preparing for subsequent actions.”

2. The authors use a significance threshold of p < 0.01. This is a plus, because threshold is smaller than in most neuroscientific studies, supporting the robustness of the findings. Nevertheless, I wonder whether there was a particular reason for choosing such a low threshold - for example, are the methods used particularly susceptible to false-positive findings?

• Response:

We thank the reviewer for this comment. We, too, believe that adopting more conservative thresholds for significance can help increase the robustness and reliability of results in neuroscience. This idea has motivated us to be more conservative and careful when assessing the significance of our results in general, and we have therefore opted to choose an alpha of 0.01 to test significance in our projects.

At the same time, research indicates that mass-univariate analysis can be more susceptible to false positives due to the high number of tests performed during analysis if adequate correction methods are not applied. This notion has further motivated us to be very conservative when assessing the significance of this particular analysis. We have revised the corresponding section in the Methods and added a motivation to justify our decision.

The section now reads: (line 323)

"This distribution was used to determine critical test values corresponding to a significance level of p = 0.01 (we used a more cons

---

## [Decision Letter · Decision Letter 1]

18 Sep 2024

Neurocognitive Dynamics of Preparatory and Adaptive Cognitive Control: Insights from Mass-Univariate and Multivariate Pattern Analysis of EEG data

PONE-D-23-34766R1

Dear Dr. García Alanis,

We’re pleased to inform you that your manuscript has been judged scientifically suitable for publication and will be formally accepted for publication once it meets all outstanding technical requirements.

Kind regards,

Yansong Li

Academic Editor

PLOS ONE

Additional Editor Comments (optional):

Review Comments to the Author

Reviewer #1: I can confirm that all my comments have been addressed sufficiently and I am happy with the current version of the manuscript.

Reviewer #2: The authors have revised the manuscript really well, thank you very much. I recommend publication of this study.

---

## [Editor Report · Acceptance letter]

30 Sep 2024

PONE-D-23-34766R1 

PLOS ONE

Dear Dr. García Alanis, 

I'm pleased to inform you that your manuscript has been deemed suitable for publication in PLOS ONE. Congratulations! Your manuscript is now being handed over to our production team.

Kind regards, 

on behalf of

Dr. Yansong Li 

Academic Editor

PLOS ONE